# Contribution of calving to frontal ablation quantified from seismic and hydroacoustic observations calibrated with lidar volume measurements

Andreas Köhler[1,2], Michał Pętlicki[3], Pierre-Marie Lefeuvre[2], Giuseppa Buscaino[4], Christopher Nuth[2], and Christian Weidle[5]

[1]NORSAR, 2007 Kjeller, Norway
[2]Department of Geosciences, University of Oslo, Post Box 1047, 0316 Oslo, Norway
[3]Centro de Estudios Científicos, Valdivia, Chile
[4]CNR, Anthropic Impacts & Sustainable Marine Environment Institute, Capo Granitola, Torretta Granitola, Italy
[5]Institute of Geosciences, Christian-Albrechts-Universität zu Kiel, Kiel, Germany

**Correspondence:** Andreas Köhler (andreas.kohler@norsar.no, andreas.koehler.geo@gmail.com)

**Abstract.** Frontal ablation contributes significantly to the mass balance of tidewater glaciers in Svalbard and can be recovered with high temporal resolution using continuous seismic records. Determination of the relative contribution of dynamic ice loss through calving to frontal ablation requires precise estimates of calving volumes at the same temporal resolution. We combine seismic and hydroacoustic observations close to the calving front of Kronebreen, a marine terminating glacier in Svalbard, with repeat lidar scanning of the glacier front. Simultaneous time-lapse photography is used to assign volumes measured from lidar scans to seismically detected calving events. Empirical models derived from signal properties such as integrated amplitude are able to replicate volumes of individual calving events and cumulative subaerial ice loss over different lidar scan intervals from seismic and hydroacoustic data alone. This enables quantification of the contribution of calving to frontal ablation, which we estimate for Kronebreen to be about 18–30%, slightly below the subaerially exposed area of the glacier front. We further develop a model calibrated for the permanent seismic station KBS at about 15 km distance from the glacier front, where 15–60% of calving events can be detected under variable noise conditions due to reduced signal amplitudes at distance. Between 2007 and 2017, we find a 5–30% contribution of calving ice blocks to frontal ablation which emphasizes the importance of underwater melting (roughly 4–9 m d$^{-1}$). This study shows the feasibility to seismically monitor not only frontal ablation rates but also its dynamic ice loss contribution continuously and at high temporal resolution.

## 1 Introduction

Glaciers are an important contributor to eustatic sea level rise in a warming climate (Gardner et al., 2013; Huss and Hock, 2015), with dynamic discharge one of the largest uncertainties in future predictions (Vaughan et al., 2013). Frontal ablation at marine-terminating glaciers comprises ice loss through subaerial and submarine melting and iceberg calving. Submarine melting strongly promotes calving through thermal undercutting (Bartholomaus et al., 2013; Luckman et al., 2015; Vallot et al., 2018; How et al., 2019). To monitor and better understand the dynamic ice loss contribution, field records of calving can

be obtained visually through human observation and time-lapse imagery (e.g., O'Neel et al., 2003; Chapuis et al., 2010; How et al., 2019), as well as through terrestrial remote sensing such as ground-based radar (Chapuis et al., 2010; Walter et al., 2019) and lidar surveys (Pętlicki and Kinnard, 2016). Calving events are also successfully detected from passive indirect techniques such as seismic (Ekström et al., 2003; Amundson et al., 2008; O'Neel et al., 2010; Walter et al., 2012; Bartholomaus et al.,
2012; Köhler et al., 2015), hydroacoustic (Glowacki et al., 2015), and water surface wave monitoring (Minowa et al., 2018).

Seismic, hydroacoustic, and water-wave methods have the advantage to produce continuous calving records with high temporal resolution and limited logistical effort. Furthermore, they are independent of visibility conditions in contrast to optical methods. However, inferring ice volumes from those signals using physical models is challenging (Podolskiy and Walter, 2016; Aster and Winberry, 2017). For example, it requires different approaches dependent on the calving style, i.e., for glacial
earthquake signals from buoyancy-driven nontabular iceberg calving such as observed in Greenland (e.g., Murray et al., 2015; Sergeant et al., 2019) or for seismic calving signals generated during iceberg–sea or lake surface interactions (Bartholomaus et al., 2012). For the first calving style, a physical model has been recently presented by Sergeant et al. (2019) who used a seismo-mechanical coupling approach to infer calving volumes from glacial earthquakes based on a catalog of contact forces computed for an iceberg capsize numerical model. The only successful approach so far for the second type of calving (sea/lake
surface impacts) is calibrating seismic or water-wave records to directly inferred calving sizes using empirical models valid for a particular glacier and recording site. For example, Bartholomaus et al. (2015) developed an empirical model to estimate iceberg volumes from individual seismic calving signals at Yahtse Glacier, Alaska, based upon visually perceived iceberg sizes (quantified on an integer scale). Minowa et al. (2019) used calving-generated tsunami signals for estimating calving volumes at Bowdoin Glacier, Greenland, where the empirical model was calibrated with ice volumes from high-resolution DEMs derived
from UAV photogrammetry. For a similar approach used at Perito Moreno glacier, Patagonia, calving sizes were estimated from calving areas visible on time-lapse camera images (Minowa et al., 2018). A different approach was applied by Köhler et al. (2016) who calibrated an empirical model of dynamic ice loss using frontal glacier ablation measured through timeseries of repeat satellite images, allowing reconstruction of frontal ablation directly from seismic calving signals with weekly resolution and going back decades. In this approach, a constant ratio between frontal melting and calving was implicitly included,
an assumption that requires validation through independent dynamic ice loss measurements.

In this study, we measure dynamic ice loss at Kronebreen, Svalbard, using seismic data recorded at three locations and precise ice volume measured from repeat lidar scanning for calibration, instead of visually perceived calving size empirically scaled to ice volume (Bartholomaus et al., 2012). In addition, we use for the first time a hydroacoustic record for calibration in addition to seismic data. Our method relies on time-lapse camera images to prepare the calibration data set. Finally, we
compare our results with independently measured frontal ablation rates at Kronebreen to assess the potential to quantify the contribution of frontal melting.

## 2    Study site and calving records

Kronebreen is a grounded, fast-flowing tidewater glacier in the northwestern part of the Arctic archipelago of Svalbard (78.88 °N, 12.55 °E) about 15 km East of the research settlement of Ny-Ålesund. The terminus is about 3 km wide with a maximum ice cliff elevation of about 60 m above and maximum depth of about 100 m below sea level (Chapuis et al., 2010; Vallot et al., 2018). It is moving with an average annual velocity of 1–3 m d$^{-1}$ (Schellenberger et al., 2015). The fjord temperatures vary seasonally between 0–7 °C (Luckman et al., 2015). Mass loss is dominated by frontal ablation (Nuth et al., 2012; Luckman et al., 2015) making it an ideal candidate to further understand the processes of calving and frontal ablation. The glacier is currently experiencing an accelerated, rapid retreat, which amounts to more than 1 km since 2012 (Schellenberger et al., 2015; Köhler et al., 2016; Vallot et al., 2018; Deschamps-Berger et al., 2019) with considerable future implications for the fjord ecosystem (Torsvik et al., 2019). At the end of August 2016 a multi-disciplinary field campaign was carried out to measure the calving front of Kronebreen continuously over a two-week period with simultaneous acquisitions from time-lapse cameras, a lidar scanner, terrestrial radar interferometry, passive seismic, and hydroacoustic arrays (Köhler et al., 2019).

### 2.1    Seismic data

Between 24 August and 2 September 2016, a temporary seismic network was deployed in the vicinity of the marine terminus of Kronebreen (Fig. 1c, Köhler et al., 2019). Eleven 4.5 Hz three-component geophones connected to Omnirecs DATA-CUBE data loggers operating with a sampling frequency of 100 Hz were arranged as two small-aperture arrays to the North (KRBN, four stations) and South (KRBS, seven stations) of the calving front of Kronebreen. The inter-station spacings of both arrays were between 120 and 780 m. During installation shallow holes were dug in the ground to accommodate the geophones. Instruments were covered with soil and the data loggers were placed at the surface. Each DATA-CUBE used two internal 1.5 V batteries as power supply which were exchanged once during the survey. We also used the continuous, long-time record of the permanent broadband Kings Bay station (KBS) in Ny-Ålesund at 15 km distance from Kronebreen which operated with a sampling frequency of 40 Hz.

### 2.2    Hydroacoustic data

The field experiment included measurements of the underwater acoustic soundscape close to the terminus of Kronebreen. Two underwater acoustic recorders (ACB and ACC, Fig. 1c) were deployed at about 45 m water depth with the hydrophones placed 10 m above the sea bottom. The acoustic data were collected from 26 August to 3 September 2016. We used autonomous recorders (SM2, Wildlife Acoustics, US) with a hydrophone recording bandwidth of 8 Hz to 150 kHz and sensitivities of −166± 1 dB re 1 V/$\mu$ Pa in the band from 100 Hz to 15 kHz. A 35 kg weight resting on the sea bottom and a small sub-surface buoy above the instrument were used to maintain the vertical alignment. The buoy was connected to the upper part of the recorder with a thin, 2 m long rope. All the components were connected with non-metallic ropes to avoid noise due to moving parts. We sampled continuously (duty cycle of 100%) with a sampling frequency of 192 kHz and a resolution of 16 bits. No pre-amplification or filtering was applied during the recordings except for the automatic anti-aliasing filter. The recording units

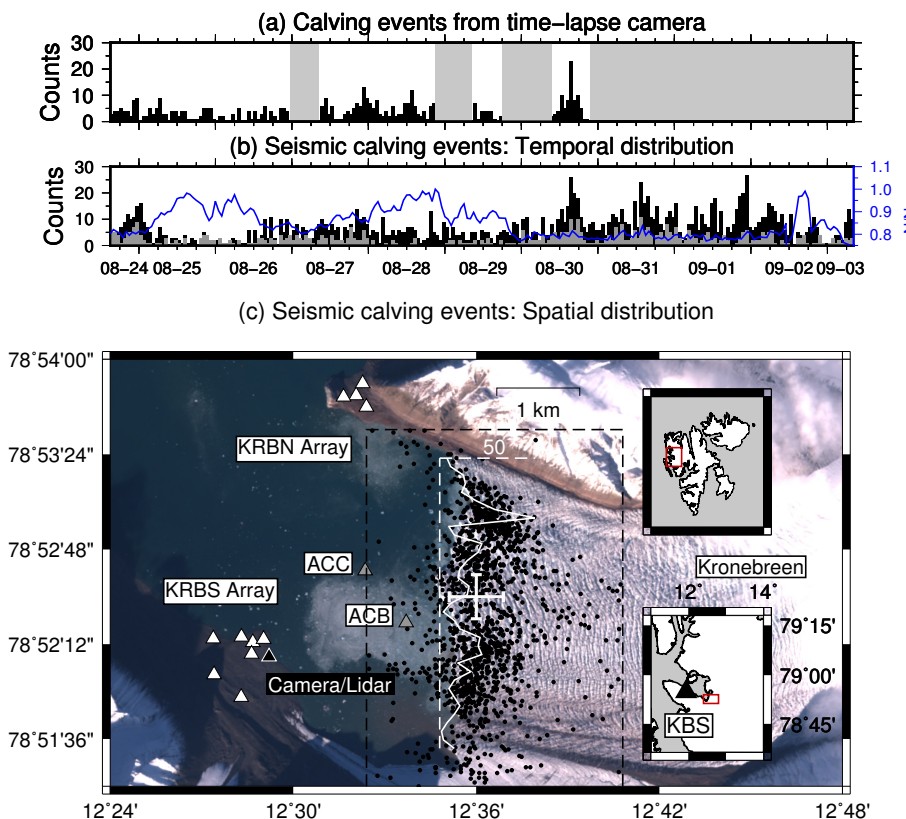

**Figure 1.** Calving event time series and locations. (a) Hourly counts of calving events automatically detected on time-lapse camera images. Gray areas indicate image gaps. (b) Hourly counts of seismic detections simultaneously at KRBN and KRBS (black) and at KBS (gray). Blue line shows relative variations in seismic background noise level (N) with respect to maximum level (Nmax) in recording period. (c) Triangles show locations of time-lapse camera (black), lidar scanner (black), seismic (white), and hydroacoustic (gray) instruments at the calving front of Kronebreen. Black points are seismic calving event locations. White error bars show an estimate of the typical location uncertainty. White curve is histogram of time-lapse camera event counts projected on a line (dashed) parallel to the terminus. Dashed box is used to classify located seismic signals as calving events. Insets to the right show Arctic archipelago of Svalbard (upper) and a closeup map of Kongsfjorden area (lower) with location of permanent broadband seismometer KBS in Ny-Ålesund.

were recovered using an acoustic releaser. The clock of the recorder was synchronized with GPS time at deployment. The clock drift measured after recovery was 10 s and was corrected accordingly assuming a linear trend.

## 2.3 Time-lapse camera image data

A time-lapse camera was installed on a tripod at the top of a moraine cone at 8 m a.s.l. (Fig. 1c). The Harbotronics set consisted of a sealed box, an intervalometer, a camera (see Table A1) and an external 12 V 30 A battery. The camera was set to take a picture every 2 seconds, but the frequency varied due to a combination of factors related to the camera, SD card, shutter speed,

battery, and temperature. The lens and focus were taped to the lowest focal length (i.e. 18 mm) in order to cover the largest part of the glacier front. The aperture was fixed to a value of 8.0 whereas the ISO and shutter speed were automatically adjusted to adapt to the ambient luminosity. The time-lapse camera was visited every one or two days to replace the SD card and correct for any clock-drift.

5     The time-lapse images are preprocessed, geo-referenced, and then used to identify calving events. For visual inspection, we extract the corresponding images within one minute around each calving event detected in the seismic record to generate an animated image sequence. Less than 5% of all events visible in the time-lapse images represent submarine calving, similar to observations at other glaciers in Svalbard (How et al., 2019). In addition, an autonomous calving event detector is implemented to compare the time series of seismicity and directly observed calving (see Fig. 1a). A detailed description of the image 10  processing and the calving detection algorithm can be found in the Appendix B (Fig. A1, Table A1).

## 2.4   Repeat lidar scanning

Volumes of individual and aggregated calving events were calculated by differencing Digital Surface Models (DSMs) obtained from repeat lidar scanning of the glacier terminus (Fig. 1c and B1). The glacier front and surroundings were surveyed with a long-range terrestrial laser scanner Riegl VZ-6000 on 23–25, 27, 30, and 31 August, and 1–3 September. On 24, 25, 27, 30, 15  and 31 August, and 1 September, a series of frequently repeated (5–30 minutes interval) lidar surveys of the ice front, restricted to the southern half of the terminus, were made. Lidar Pulse Repetition Rate (PRR) was always set to 50 kHz and horizontal and vertical angular resolution was typically set to $\sim 0.004\,°$. The lidar signal reflected from the northern part of the terminus was too weak to provide a reliable DSM due to large distance and, additionally, it was partially occluded from view by the protruding portion of the ice cliff near the centerline.

20     Resulting point clouds were preprocessed in Riegl RiSCAN-PRO software. Geocoding was made with a Multi Station Adjustment (MSA) plugin using ice-free areas of lidar scans acquired from two auxiliary scan positions: one located near the KRBN array and a second to the west of the main scan position. Lidar positions were measured with static observations of differential GPS (Leica GS14) using NYA1 as a reference station. Subsequently, all the point clouds were aligned to a common reference frame using an iterative closest point (ICP) algorithm. Multiple cloud alignment yielded mean global positioning error 25  of 8 cm. Each resulting point cloud was cropped to the ice cliff face by manual removal of the points related to the floating icebergs and boats. Point clouds were processed with Cloud Compare software in which horizontal mismatches between sequential point clouds were removed using the M3C2 plugin (Lague et al., 2013). Results were projected to a new gridded point cloud with a fixed spatial resolution of 10 cm. Then, it was separated into two classes based on the sign of the distance change between sequential point clouds: ice terminus was classified as affected by calving when the change was positive, i.e. 30  the distance from lidar scanner to the ice front increased due to calving; where the sign was negative, ice front advanced. Sub-clouds classified as affected by calving were segmented to particular calving events with the method of Awrangjeb (2016) that employs maximum point-to-point distance in the input data to classify boundary edges. Calving event scars have low reflectance in near infrared part of spectrum used by lidar (Podgórski et al., 2018) and this causes a decrease in point cloud density. Therefore, total volume of a calving event was calculated as a product of calving scar area and average calving event

depth (Fig. B1). Owing to the short time between subsequent lidar acquisitions (<1 hour), ice advection was neglected as front advance during such time differences is much lower (1–6 cm) than calving event depth (meters). For inter-daily point cloud differencing, front advance related to ice advection that takes place between measurements cannot be neglected. Hence, in such case the approach of Pętlicki and Kinnard (2016) was used, where point cloud differences in adjacent areas not affected by calving (ice advection) are added to the displacement caused by calving. Laser penetration into the glacier ice (less than 10 cm) could introduce a systematic error throughout the dataset which would, however, be canceled out during point cloud differencing and does therefore not influence the result. Measurement uncertainties are shown in Fig. 5, Table 1, and Fig. E1.

## 3  Detection, location, and validation of calving signals

Seismic arrays allow detecting weak seismic signals and suppressing uncorrelated noise by beamforming (Schweitzer et al., 2012). Frequency-Wavenumber (FK) analysis for example allows measuring the direction towards the signal source (back-azimuth) through maximizing the stacked, travel-time corrected spectral amplitudes, the so called absolute beampower. Array analysis is in particular useful for calving events at Kronebreen, since these signals do not exhibit clear P and S wave onsets required for travel-time-based epicenter inversion (Köhler et al., 2015).

We compute a time series of absolute FK beampowers in 1 s long time windows with 50% overlap for each array between 1 and 3 Hz, the predominant frequency band of seismic calving signals at Kronebreen (Köhler et al., 2015). Subsequently, a short-term over long-term average (STA/LTA) trigger (Withers et al., 1998) is applied independently to both beampower time series (STA=2 s, LTA=20 s, threshold=6). An event is declared if the STA/LTA threshold is exceeded on both arrays. The STA/LTA parameters are found through visual evaluation of selected seismogram time windows. The detections are then located using the spatial mapping by multi-array beamforming method (SMAB, in supplementary information in Köhler et al., 2016). The SMAB method assumes a straight ray path from the source to each array stations, a condition we find not to be sufficiently fulfilled, most likely due to 2D propagation effects along the ice–ocean–land interfaces. We therefore empirically correct for the azimuthal bias observed at each array during SMAB processing (see Appendix D and Fig. D1 for details).

The long wavelength of seismic calving signals ($\lambda = 400–1000$ m) limits the spatial resolution to approximately 100–250 m ($\lambda/4$, Rayleigh criterion). Furthermore, seismic propagation effects not explained by the azimuthal correction can contribute to the overall location uncertainty. A conservative estimate of the average uncertainty is 320 m (longitude) and 260 m (latitude) (Fig. 1c) which are the medians of the difference between individual SMAB locations (longitude / latitude coordinates) and directly observed calving location from the time-lapse imagery (see scatter plot Fig. D1c). For further processing, we select all events with normalized SMAB beampower >0.5 and with source locations inside a box around the terminus of Kronebreen (box dimension: $12.54°E < lon < 12.68°E$, $78.848°N < lat < 78.8926°N$, see Fig. 1c). Comparison with time-lapse images shows that a few calving events are detected as two or more separated seismic signals. Therefore, consecutive signals in our catalog are merged in a post-processing step if seismic locations are less than 400 m away from each other along the calving front and the detection time difference is less than 10 s.

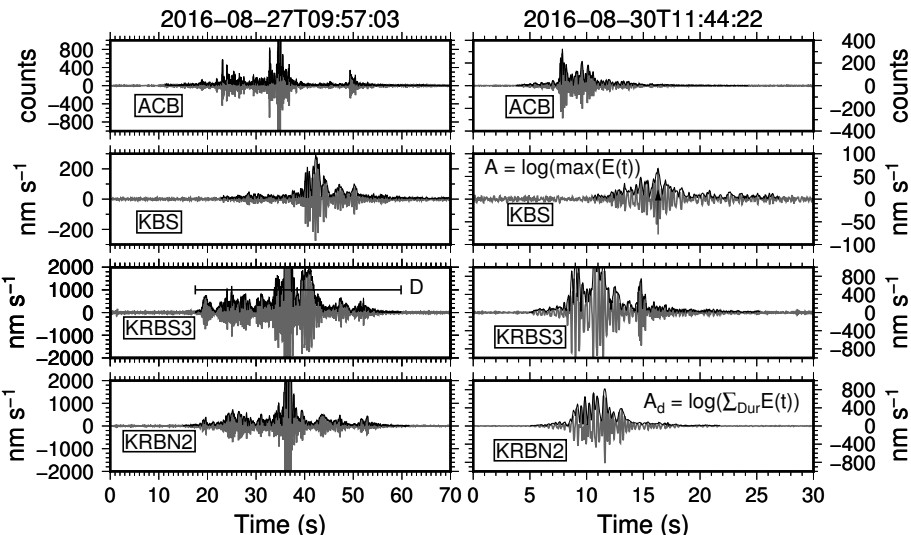

**Figure 2.** Examples of hydroacoustic (ACB) and seismic (KBS, KRBN2, KRBS3) signals of two calving events. Waveforms bandpass-filtered between 2 and 12 Hz are shown in gray. Black curves are signal envelopes ($E(t)$) used to compute the event features: signal duration ($D$), logarithm of envelope amplitude integrated over event duration ($A_d$), and logarithm of maximum envelope amplitude ($A$). Note, that for hydroacoustic data the raw digitizer output is processed (counts).

The seismic record of the permanent station KBS has been previously used to generate a catalog of calving events between 2001 and 2015 (Köhler et al., 2016). For this study, we updated this catalog by processing data of recent years (2016–2017) using the same detection method. The calving event classifier for KBS was retrained using seismic signals from 2016 confirmed to originate at Kronebreen using our temporary seismic deployment and the time-lapse images. KBS detections during the field
experiment are shown in Fig. 1b (gray histogram).

In total 1,460 seismic calving event signals are simultaneously detected on array KRBN and KRBS within the measurement period. As expected, the detection rate is affected by the seismic background noise level, i.e., less events are being detected during windy time periods when, for example, ocean waves generate more ambient vibrations (Fig. 1b). The spatial distribution (Fig. 1c) suggests that calving is not uniformly distributed along the terminus. The majority of seismic detections are visually
confirmed on time-lapse camera images. Visual inspection of time-lapse images for all events on 30 August reveals that only 4% of all seismic detections located within the camera-covered range of the terminus cannot be visually confirmed (false positives). Most non-confirmed signals (16% of the total number of detections) originate from outside the camera range at the southern end of the terminus, or are concealed from the observer as a result of the front geometry. We also assess the possibility of missed calving events (false negatives) using the catalog of events independently obtained from time-lapse camera images (see
Appendix B and Fig. 1a). Only 7% of all events caught on camera cannot be matched with a simultaneous seismic detection on both arrays. However, weak seismic signals can be identified on one of both arrays for many of those events. Since we require detection on both arrays for a more accurate location, we do not include these events in further processing. Days of

high seismic noise level (25 and 28 August) account for more than 50% of these false negatives. Depending on the noise level, between 10 and 45% of all calving signals observed on the arrays at Kronebreen are detected at the permanent station KBS, consistent with an earlier experiment conducted in 2013 (Köhler et al., 2016).

## 4  Model development, calibration, and results

To find a relation between seismic calving signals and measured ice volumes, we compute the following features for each detection after bandpass-filtering between 1.5 and 5 Hz (Bartholomaus et al., 2015; Köhler et al., 2016): signal duration ($D$), logarithm of envelope amplitude integrated over event duration ($A_d$), and logarithm of maximum envelope amplitude ($A$). Detailed information about how the duration is determined is given in the Appendix C. After testing all sensor records, we decided to use a single station of each array exhibiting the best signal-to-noise ratio (KRBN2, KRBS3). Furthermore, we use

the seismic detection times to extract the corresponding calving signals from the record of the hydrophone located closest to the calving front (ACB, Fig. 1c) and compute the same features as for the seismic signals. Although hydroacoustic data provide a much broader frequency range, we find the computation of features below 15 Hz to be most robust for calibration to ice volumes. The reasons are underwater noise level being in general higher and calving-related signals tending to be less well-defined and localized in time at higher frequencies. We find that hydroacoustic calving signals often consist of multiple

arrivals and weak signals emitted in the aftermath of the ice–water impact, possible caused by ice avalanches at the terminus, ice breakup at the calved ice block, and air bubble noise from melting freshly calved ice (Urick, 1971; Tegowski et al., 2014; Pettit et al., 2015). Figure 2 presents signals of two typical calving events on KRBN2, KRBS3, ACB, and KBS with signal envelopes shown as black curves.

From repeated lidar scans with 30 minute intervals at daytime on 24, 25, 27, 30, and 31 August, and 1 September, we obtain

100 individual calving volumes. Three sources of uncertainty exist when matching these volumes with the corresponding seismic and hydroacoustic calving signals: (1) The lack of exact timing in the lidar catalog, i.e., we only know that the event occurred within the scanning interval, (2) the assumption that a volume measured at a particular location corresponded to a single event, not to multiple events spread out within the scanning interval, and (3) the seismic location uncertainty. In other words, we have accurate timing but uncertain locations in case of seismic observations, and uncertain timing but precise

locations for events in the lidar catalog. The time-lapse camera images are thus used to manually identify as many high-confidence matches between the seismic and the lidar catalog as possible. For each seismic calving event, the corresponding camera image series is inspected, and the location of the calving event at the terminus is visually identified. We then select all events from the lidar catalog during the scanning interval including the seismic signal at this location. Doing so, we take into account the location of the laser scan focus at the seismic detection time. Lack of continuous images due to camera failure

on 27 and 31 August, and 1 September reduces the number of usable lidar volumes to 88, from which we are only able to unambiguously match 35 events. Unsuccessful matching is mostly due to events that occurred outside the camera range at the southernmost part of the terminus or due to the presence of too many candidates for matching. In a few cases, several isolated volumes with close proximity to each other correspond to a single calving event, i.e., to a single seismic signal. These volumes

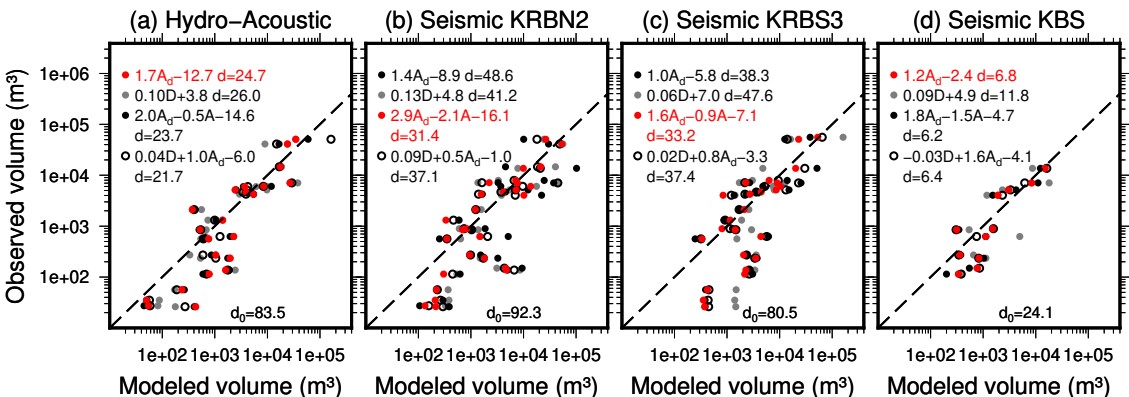

**Figure 3.** Generalized Linear Models (GLMs) relating waveform properties to calving volume (lidar scanning) of individual events. Models are calibrated for different stations and combination of calving signal features ($D$: signal duration, $A_d$: amplitude integrated over event duration, $A$: maximum amplitude). Model coefficients, deviance ($d$), and Null deviance ($d_0$) of each model are stated. Red symbols show the best-performing models. Dashed lines indicate equality of modeled and observed volumes.

are merged accordingly. Furthermore, several calving events can occur at the same location during the scanning interval. In this case, we merge the corresponding seismic signals and treat them as a single event. This merging is done iteratively. We first calibrate signal properties to ice volumes excluding those signals and then re-calibrate our regression model using signal features merged according to the preliminary model (see below for more details). The final number of calving events available

for calibration is 27 for the seismic and 22 for the hydroacoustic data. Since only the largest calving events are observed at 15 km distance, this number reduces to 12 events for the KBS record.

Following the approach of Bartholomaus et al. (2015), we use Generalized Linear Models (GLMs, McCullagh and Nelder, 1989) and test different combinations of features computed from the seismic or hydroacoustic calving signals to find an empirical relation to estimate the lidar calving volumes. The linear combination of predictor variables (seismic or hydroacoustic

features) is related to the response variable (calving volume $V$) through a logarithmic link function. As in Bartholomaus et al. (2015) and Köhler et al. (2016) we employ the Gamma distribution to model the response variable. The GLMs have the form:

$$\log(V) = c_1 \cdot S_i + c_2 \cdot S_j + c_0, \tag{1}$$

where $c_{0,1,2}$ are the time-invariant model coefficients and $S_i/S_j$ are chosen from the features $D$, $A_d$, and $A$ computed from the hydroacoustic (ACB) and seismic calving signals (KRBN2, KRBS3, and KBS). Merging $k$ multiple signal features for a single volume measurement as mentioned above is done by computing:

$$S_{merged} = \left( \log\left( \sum_k \exp(c_1 \cdot S_k + c_0) \right) - c_0 \right) / c_1, \tag{2}$$

where $c_0$ and $c_1$ are obtained from a preliminary model trained using a single predictor variable (the feature to be merged) and without using the to-be-merged calving signals. Subsequently, the GLM is re-estimated including merged signals.

Figure 3 shows the results for 16 models using a single ($c_2=0$) or two predictor variables (see also Fig. E1). The GLM performance is evaluated using the deviance $d$, the common quality-of-fit statistic for GLMs (McCullagh and Nelder, 1989). The Null deviance $d_0$ stated in Fig. 3 is the statistic for regression using a single model parameter $c_0$, and is equal for all models for a given station. In addition, the significance of model coefficients $c_1$ and $c_2$ to be non-zero is assessed using a T-test with 90% confidence. Finally, we perform a cross-validation test where we divide the data set randomly into 80% training and 20% validation data. This is repeated 500 times, and the mean of all RMS errors is computed.

Using deviance, cross-validation RMS errors, and T-test, we identify one or two best-performing models for each station (highlighted in red in Fig. 3). The results show that $A_d$ alone or in combination with $A$ tends to perform best for all stations. In case of the seismic data, combining $A_d$ and $A$ results in better models than using a single variable, with model coefficient $c_2$ for $A$ always being negative. An intuitive explanation for this result is that $A$ removes the effect of short-time amplitude spikes in the calving signal from the quantity $A_d$. These spikes might be caused by secondary processes during calving that are not directly related to the impact of the total ice volume into the water (e.g., ice projectiles, effects of the geometry of ice block, etc.).

Scatter between observed and predicted event volumes could be related to different factors affecting seismic and acoustics signals properties, for example source effects, such as the type of calving, shape of ice block, and/or manner of impact, as well as wave propagation effects depending on the travel path from different locations at the calving front to the receiver. We find that model performance does not improve significantly after including proxies for these effects. For example, multiplying $A_d$ and $A$ with the square root of the source distance to account for geometrical spreading (assuming surface waves), modeling the potential energy of the calved ice block instead of the ice volume to take into account fall height, or including the seismic/acoustic noise level as an additional predictor variable since it can affect the measured length of a calving signal, does not improve the model fit.

## 5 Model validation with cumulative ice loss from lidar scans

Our results clearly show the existence of an empirical relation between calving signals properties and logarithm of ice volumes approximated by a linear trend. To validate our empirical models calibrated with individual, successfully matched calving events (i.e., a subset of all lidar ice volumes), we employ the cumulative subaerial ice loss measured from lidar scanning within different time periods as a test data set. The sum of volumes over all scans during daytime on individual days as well as volumes obtained from the inter-daily scans are used.

First, we apply the best empirical models of each station obtained from the full calibration data (Fig. 3) to estimate the cumulative calving volume using all calving signals observed during the measurement period from 24 August until 2 September (white symbols in Fig. 4). In a second step, in order to estimate its uncertainty, we randomly select 80% of the calibration data set to determine a new GLM $M_{cv}$. Then, the cumulative calving volume $V_{cv}$ of all observed calving events is computed using $M_{cv}$. This is repeated 100 times. Figure 4 shows the resulting histograms plots for $V_{cv}$ for all sensors. The distribution gives an impression about the uncertainty of the estimated total ice loss. Since the hydroacoustic recording period is shorter, we repeat

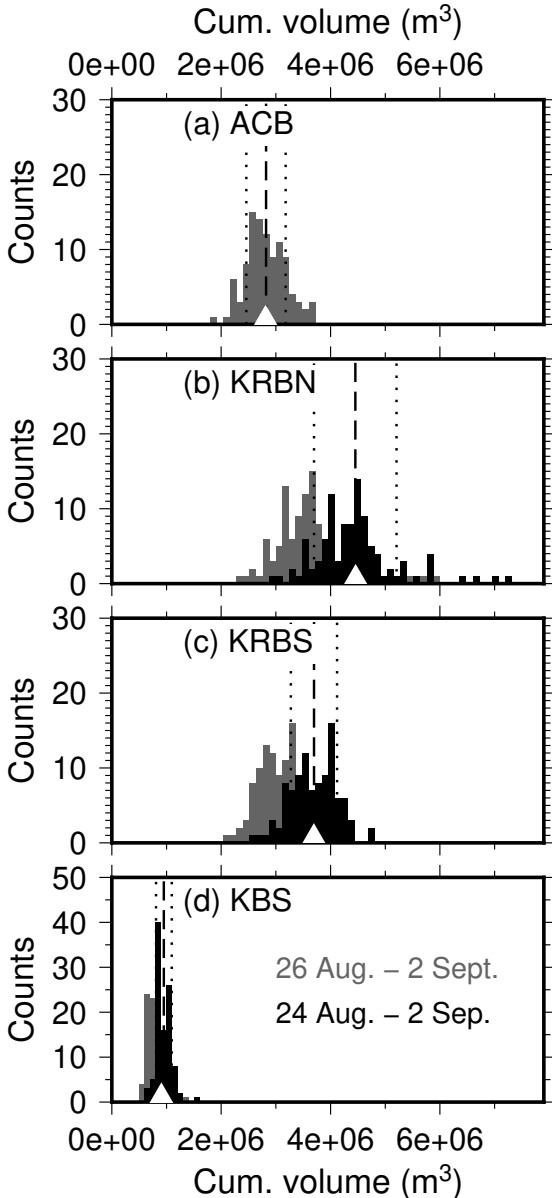

**Figure 4.** Modeled distribution of cumulative ice volumes between 24 August and 2 September using best GLMs in Figure 3 that are trained with 80% (randomly selected) of the calibration data in 100 runs (black histograms). White symbol is volume obtained with model calibrated with full data set. Gray histograms correspond to hydroacoustic recording period (from 26 August). Vertical lines indicate means (dashed) and standard deviation (dotted) obtained from the volume distribution.

this processing for the seismic data within the corresponding time period (gray histograms). Assuming a normal distribution

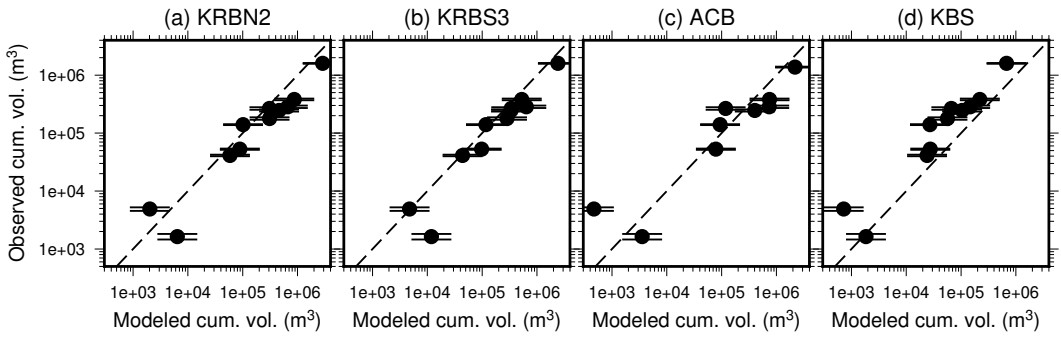

**Figure 5.** Modeled vs. observed cumulative volume for different lidar scanning intervals and stations in 2016 calibrating period. Lidar measurement uncertainties are indicated with error bars (vertical distance between caps; too small to see for high volumes).

|  | Measurement | Obtained from seismic/hydroacoustic records with empirical models | | | | |
| Interval | Lidar | KRBN2 | KRBS3 | ACB | KBS | FA |
| --- | --- | --- | --- | --- | --- | --- |
| 25 Aug daytime | 41.19±1.2 | 58.58 | 43.92 | - | 23.96 | - |
| 27 Aug daytime | 52.78±1.2 | 88.66 | 97.72 | 78.72 | 27.50 | - |
| 30 Aug daytime | 1.64±0.2 | 6.41 | 11.90 | 3.56 | 1.84 | - |
| 31 Aug daytime | 4.89±0.3 | 2.03 | 4.74 | 0.47 | 0.72 | - |
| 1 Sep daytime | 139.87±2.4 | 101.36 | 117.22 | 93.88 | 27.01 | - |
| 24–25 Aug | 178.42±8.5 | 309.61 | 284.24 | - | 56.39 | - |
| 25–27 Aug | 265.76±13.3 | 307.01 | 338.60 | 118.17 [1] | 66.46 | - |
| 27–30 Aug | 285.03±13.5 | 665.77 | 640.05 | 737.20 | 146.58 | - |
| 30–31 Aug | 381.29±12.5 | 872.13 | 528.40 | 750.04 | 220.39 | - |
| 31 Aug - 1 Sep | 245.75±10.0 | 460.57 | 336.69 | 402.75 | 104.41 | - |
| 24 Aug - 2 Sep [2] | 2,737.12 | 4,500±800 | 3,700±400 | 2,800±400 [1] | 950±140 | 18,000±2,000 |
| 2016 | - | - | - | - | 28,000±5,000 | 640,000±80,000 |

**Table 1.** Cumulative dynamic ice loss modeled and measured (lidar) for different time intervals in units of $10^3$ m$^3$. Only calving signals originated in the scanned range of terminus are used. FA: Frontal ablation (including submarine calving and melt) obtained from Köhler et al. (2016). [1] Acoustic data available from 26 Aug. [2] Lidar volume until 3 Sep and using all calving signals (entire front).

for $V_{cv}$, the means and standard deviations are computed for each station to quantity the uncertainty of the total calving volume estimate (Table 1).

Using all seismic calving observations at KRBN and KRBS during the field experiment, the total ice volume lays consistently for both models between $2 \cdot 10^6$ and $6 \cdot 10^6$ m$^3$ (Fig. 4, Table 1) which corresponds to an average daily ice loss rate of 0.22–0.67·$10^6$ m$^3$d$^{-1}$. Furthermore, $V_{cv}$ values from the hydroacoustic data are in agreement with seismic data in the corresponding time period (Fig. 4). The volumes obtained from KBS lay between $0.5 \cdot 10^6$ and $1.5 \cdot 10^6$ m$^3$, suggesting that calving observed seismically at a distance of 15 km only accounts for about 25% of the ice loss estimated close to the glacier. For comparison,

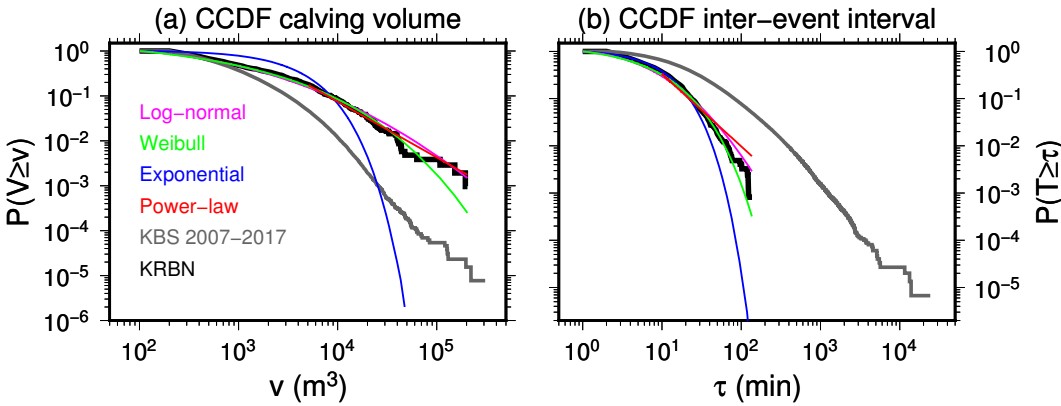

**Figure 6.** Complementary cumulative distribution functions (CCDF) for calving event volume (v) estimated with our model from seismic signals and observed inter-event interval ($\tau$). Different distribution models are fitted to the data (KRBN only). Loglikelihood ratio (positive value favors first model) and their significance for two candidate distributions (power-law=P, exponential=E, log-normal=L, Weibull=W): (a) P/E: 41.8, 0.017; P/L: -1.4, 0.35; P/W -1.4, 0.34; L/E: 43.2, 0.008; L/W: -0.0017, 0.98; E/W: -43.2, 0.00; (b): P/E: -2.0,0.85; P/L: -15.8, 0.001; P/W: -15.0, 0.0007; L/E: 13.8, 0.02; L/W: 0.8, 0.3; E/W: -13.0, 0.00.

the ice loss at Kronebreen measured with lidar scanning between 24 August and 3 September is $2.7 \cdot 10^6$ m$^3$. Since parts of the calving front were not scanned, the actual volume is expected to be higher, which is consistent with the higher values obtained from our GLMs. Figure 5 shows a comparison between modeled and observed cumulative volumes for different temporal lidar scanning intervals considering only seismic/hydroacoustic signals that occurred inside the actually scanned region (see also

Table 1). There is a good correspondence with respect to the magnitude of obtained volumes. However, especially for interdaily time intervals, our models tend to overestimate the ice loss, which reflects the general model uncertainty. As expected, ice volumes are underestimated at KBS since only a fraction of calving signals is observed at larger distance.

The complementary cumulative distribution functions (CCDF) of estimated calving volumes and observed inter-event intervals are shown in Fig. 6 for KRBN and KBS (2007–2017). Relative contribution of longer inter-event intervals at KBS is

10 higher than at KRBN due to the catalog incompleteness, i.e., weak calving signals undetected at larger distances. The difference in volume CCDFs at both stations is less easily explained. While lacking low-volume event detections at KBS would lead to higher probabilities to observe larger volumes, we observe in fact the opposite. The reason is most likely that we tend to underestimate high-volume events at KBS with our model since the corresponding signals are not well represented in our calibration data set at that station. Hence, we have to keep in mind that the observed underestimation of total calving ice loss at

15 KBS mentioned above is affected by two factors: catalog incompleteness as well as worse model performance for large calving events. We try to fit the CCDFs at KRBN using different distribution models (Fig. 6) by comparing two model fits using the loglikelihood ratio and its significance (Alstott et al., 2014). In agreement with previous studies (Chapuis and Tetzlaff, 2014; Pętlicki and Kinnard, 2016), calving event size distribution can be best fitted by a log-normal or power-law distribution, the latter one only for volumes larger than 5000 m$^3$. In contrast to Minowa et al. (2019), an exponential model does not explain

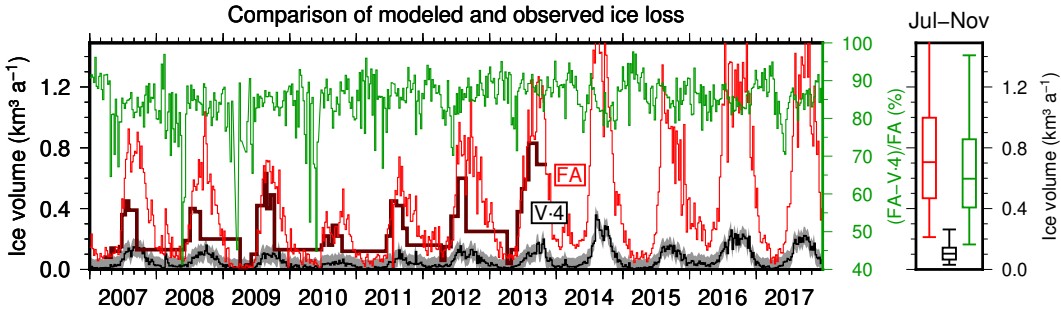

**Figure 7.** Comparison of frontal ablation (FA) rates at Kronebreen directly measured with satellite remote sensing between 2007 and 2013 (dark red) and modeled in Köhler et al. (2016) from seismic KBS data with weekly resolution (red). Scaled dynamic ice loss rates (V·4) modeled in this study from KBS record and taking into account an average dynamic ice loss detection rate of 25% at KBS are shown in black with gray area indicating uncertainty (two times standard deviation estimated in 2016 calibration period, Table 1). Green curve is percentage of total frontal ablation not explained by modeled dynamic ice loss. Right panel shows box plots for calving season (2%, 25%, 98%, and 75% quantiles) using difference (FA-4·V) instead of percentage.

the observed calving size distribution at Kronebreen. We also confirm that inter-event intervals do not seem to follow exponential or power-law models (Chapuis and Tetzlaff, 2014; Pętlicki and Kinnard, 2016), and that using Weibull and log-normal distribution results in better fits.

## 6   Comparing dynamic ice loss and total frontal ablation

In Fig. 7 we compare cumulative weekly dynamic ice loss obtained from our models with the total frontal ablation (which includes frontal melting) measured directly with satellite remote sensing data between 2007 and 2013 as well as modeled from the seismic KBS record between 2007 and 2017 (Köhler et al., 2016). A pronounced increase in modeled calving ice loss after 2013 is apparent, coinciding with the recent accelerated retreat of Kronebreen (Deschamps-Berger et al., 2019). Our results show an increase of dynamic ice loss rates, and also suggest longer calving seasons in recent years. The annual frontal ablation

had values typically between 200 and 500·$10^6$ m$^3$ within the past 15 years (Köhler et al., 2016; Schellenberger et al., 2015). Dynamic ice loss modeled from KBS data in this study for the same time period shows as expected much lower rates (Fig. 7). As shown in the previous section, only a fraction of all subaerial calving events is observed at KBS due to significant seismic amplitude attenuation at 15 km distance. The percentage of detected events varies over time since noise level and possibly also the relative contribution of small vs. large calving events may change. Within our calibration period the variation

of completeness causes the modeled ice loss to vary between 15 and 58% of the measured values (see Table 1). Assuming an ice loss detection rate of 25%, comparison between ice loss modeled in this study and frontal ablation estimated in Köhler et al. (2016) suggests that between 70 and 95% of the frontal ablation does not occur through the physical process of iceberg calving (green curve in Fig. 7).

For the time period of our experiment between 24 August and 2 September, the approach of Köhler et al. (2016) yields an ice loss of about $18 \cdot 10^6 \, \mathrm{m}^3$. Since about two-thirds of the grounded vertical terminus area of Kronebreen are submarine (Lindbäck et al., 2018) and assuming that submarine ice loss occurs mainly through melting as suggested by our visual calving observations at Kronebreen and How et al. (2019), the corresponding dynamic ice loss would be $6 \cdot 10^6 \, \mathrm{m}^3$ which is just slightly above the values estimated from the KRBN and KRBS records in this study, i.e., the contribution of calving to frontal ablation would be 18–30% (Table 1). For the entire year of 2016, the approach of Köhler et al. (2016) yields an ice loss of about $640 \pm 80 \cdot 10^6 \, \mathrm{m}^3$. Taking again into account varying seismic catalog completeness at KBS resulting in underestimation of calving volumes (15–58% of true ice loss) and assuming that only subaerial ice loss contributes (about 33%), we would expect to obtained with our model for KBS between 5 and 20% of the actual frontal ablation, i.e., values within a range of 30 to $130 \cdot 10^6 \, \mathrm{m}^3$. In fact, we obtain $28 \pm 5 \cdot 10^6 \, \mathrm{m}^3$ in this study. Being located at the lower bound of the expected range is consistent with the contribution of dynamic ice loss to frontal ablation estimated above and might be related to a lower calving catalog completeness during the fall, when calving usually continues with high activity, but frequent storms increase the amplitudes of ambient seismic noise in the Kongsfjord area. Furthermore, the model of Köhler et al. (2016) was calibrated for the time period 2007–2013 which could result in a higher uncertainty or a bias in 2016.

## 7 Discussion

A major novelty of our approach, estimating iceberg volume directly from seismic or hydroacoustic calving signals, is that a source of uncertainty in the empirical model is removed. Instead of qualitatively estimating the relation between visually perceived calving sizes in the field and the actual volume as previous studies did, we use precise lidar scanning of the calving front for calibration. Our results clearly show that this approach works. The choice of the regression method (GLM) and calving signal features is similar and builds upon to the method presented by Bartholomaus et al. (2015). We confirm that the length of the signal represents the ice volume better than the maximum amplitude (Qamar, 1988; O'Neel et al., 2007b; Bartholomaus et al., 2015; Köhler et al., 2016). However, in contrast to previous studies, we find that empirical models using the integrated amplitude perform better than using the signal duration only. The difference of our approach compared to estimating frontal ablation from seismic calving records in Köhler et al. (2016) is a higher temporal resolution and the focus only on dynamic ice loss through modeling volumes of individual calving events. Therefore, our models do not use a time-normalized sum of seismic signal properties and the catalog incompleteness in specified time periods as predictor variables as done in Köhler et al. (2016).

### 7.1 Applicability and opportunities of seismic calving quantification

By calibrating a model for the permanent seismic station KBS in the vicinity of Kronebreen, our method allows a unique quantitative long-term monitoring of calving which is necessary to predict how glaciers will respond in a warming climate. Given that calibration experiments have been carried out, this approach can be applied to other similar situations worldwide where permanent seismic stations have been installed in the vicinity of a tidewater glacier. In our case, small and moderate-

sized calving events are not detected and the long-time event catalog might be contaminated by calving signals from other glaciers in the region because KBS is located at 15 km distance and using a single station limits location accuracy. However, our results for the short-time record show that our calving quantification approach benefits from instruments deployed closer to the terminus and/or small-scale seismic arrays that allow better location and detection of weak calving signals.

Frontal ablation can be directly measured through satellite remote sensing or obtained from seismic records with empirical models and weekly resolution (Köhler et al., 2016). Combining these observations with dynamic ice loss estimates as presented in this study offers the potential to investigate the contribution of frontal melting to total frontal ablation. For example, we can compute an estimate for ice loss through melting from our modeled dynamic ice loss and frontal ablation at Kronebreen by taking into account the (constant assumed) seismic catalog incompleteness at station KBS (Fig. 7). The magnitude and temporal

variability of this non-dynamic ice loss, however, is affected not only by variation of frontal melt, but also by changing catalog completeness, i.e., more weak signals remain undetected if background noise increases, which in theory can also include changing contribution of undetected submarine calving. Hence, variations in the computed contribution of the frontal melting proxy have to be interpreted with caution. Using the method to asses to calving catalog incompleteness at KBS suggested by Köhler et al. (2016) would in theory allow distinguishing between variations in calving and variations in noise. This would

be a major step forward in better understanding processes at ice–ocean interfaces. Deploying instrument arrays close to the glacier terminus, as done during our experiment in 2016, but for long-term monitoring makes seismic/hydroacoustic catalog completeness less an issue. Nevertheless, apart from limited capability to asses its temporal variability, we are able to give an estimate for the average contribution of dynamic ice loss to frontal ablation at Kronebreen of 5–30% (Fig. 7, right panel). Similar values were found recently at other tidewater glaciers (Minowa et al., 2019; Walter et al., 2019). This implies and

confirms recent studies that water temperature is the main driver of frontal ablation through underwater melting and thermal undercutting which in consequence controls subaerial calving (Bartholomaus et al., 2013; Luckman et al., 2015; Vallot et al., 2018; How et al., 2019; Mercenier et al., 2019). Using the cross-sectional submarine area (0.23 and 0.29 km$^2$ in 2009 and 2014, respectively), our residual frontal melting component of Kronebreen of 0.4–0.8 km$^3$ a$^{-1}$ (Fig. 7) would imply a retreat rate of 4–9 m d$^{-1}$ due to these processes alone. This is reasonable given the front position behavior and horizontal velocity measured

over the past decade (Schellenberger et al., 2015; Luckman et al., 2015). Schild et al. (2018) reported melt rates of 0.1–6.8 m d$^{-1}$ between April and October 2016 which is of the same order as our average rate estimate for the whole time period of 11 years. The modeled melt rate at Kronebreen according to Holmes et al. (2019) varies seasonally between 0.5 and 2.0 m$^3$ s$^{-1}$ (0.015-0.06 km$^3$ a$^{-1}$), which is one order of magnitude lower than our estimate. However, Holmes et al. (2019) point out that melt rates are probably underestimated since they are modeled from fjord temperatures and do not consider melting

from existing plumes. Therefore, the melt rates only account for 25.6% of the frontal ablation which is much lower than the contribution of more than 70% suggested by this study.

   We calibrate our models with subaerial calving events. It is well-known that also submarine calving generates underwater acoustic and weak seismic signals as well as water surface waves (Minowa et al., 2018). Glowacki et al. (2015) showed that hydroacoustic calving signals at Hansbreen, southern Svalbard, allow distinguishing submarine and subaerial calving events

based on spectral characteristics. However, at Kronebreen we found it difficult to distinguish the seismic calving signals from

subaerial and submarine calving. Seismic signals from submarine events are in general weaker, but are often indistinguishable from weak subaerial calving events or occur simultaneously with those. Hence, calving at the submarine part of the terminus is partially included in our modeled ice volumes, but could be underestimated if we assume the corresponding low-amplitude signals to be subaerial. This is similar to the effect of increasing seismic background noise level which results in an apparent

shorter duration of the calving signal. However, we observe only a few percent of all calving events to be submarine in the time-lapse images, and therefore have good evidence that our ice loss estimates are not significantly affected. This is supported by recent observations made by Minowa et al. (2018) who found 98% of all calving events occurring subaerially as well as by How et al. (2019) at Tunabreen, a tidewater glacier in central Svalbard, where the value was 97%, even though 60–70% of the terminus is below sea level, similar to Kronebreen. Nevertheless, future studies should use our data set to more systematically

explore how hydroacoustic calving signals differ for submarine and subaerial calving events at Kronebreen, and how they can be used to better understand submarine calving, similar to the approach carried out at Hansbreen (Glowacki et al., 2015).

## 7.2   Potential model extensions

Being limited by a low number of calibration data, seismic and lidar records acquired over a longer time period will offer the potential to further improve our empirical models by reducing prediction uncertainty. Since we encountered challenges

related to matching lidar volumes and calving signals when calving within the scanning interval occurs frequently, we will scan multiple smaller areas in future field experiments to unambiguously relate more calving events and their signals. This was already tested during the last days of our experiment using a single scanner and a repetition rate up to 5 min. However, the measurement period was not long enough to capture a sufficiently high number of calving events. Another possible solution is to derive ice volume from a multiview structure-from-motion photogrammetry with a multiple time-lapse camera array

(Mallalieu et al., 2017). This method allows for higher temporal resolution and is much more economic than lidar scanning, however, it requires a favorable geometry setup that is hard to meet for tidewater glaciers. Using UAV-derived DEMs (Minowa et al., 2019) is not an alternative for Kronebreen, but can be an option in case of infrequent calving when a single calving event can be clearly identified between two DEMs. Furthermore, it was shown recently that terrestrial radar interferometry can be used to measure calving sizes (Walter et al., 2019). Another approach which we tested and which provides more calibration data

is to measure calving areas from time-lapse images extracted at the seismic detection times and then scale to volumes using an empirical relation between camera-measured area and volume found by regression using the available lidar data. Alternatively, the area-volume relation for calving suggested in the literature can be used (Pętlicki and Kinnard, 2016; Minowa et al., 2018). However, similar to the method using visually quantified calving volumes, two empirical models have to be combined which results in a larger prediction uncertainty.

Our calving volume estimates of individual events include uncertainty since the calibration data set is limited and not all parameters affecting the calving signals can be modeled yet in the regression. The interaction between ice and water during calving is complex, and it is therefore not surprising that signal features have limited predictor ability for the volume. Ultimately, physical models such as proposed by Bartholomaus et al. (2012) or numerical modeling of the ice–water impact would be required to better understand seismic/hydroacoustic signal generation. Furthermore, our empirical models are valid

for a particular calving style mainly observed in Svalbard, Alaska, and Patagonia where most seismic calving signals are generated by the impact of the iceberg in the fjord (Bartholomaus et al., 2012). In contrast, the glacier mass loss through calving in Greenland and Antarctica is dominated by the breakup of large tabular or nontabular icebergs, respectively, representing different types of seismic source mechanisms (Chen et al., 2011; Murray et al., 2015). Determination of calving sizes from those seismic calving signals (glacier earthquakes) therefore requires a different approach (Sergeant et al., 2019).

Our models are calibrated for a particular glacier, terminus environment, and seismic/hydroacoustic sensor setup. Hence, application to seismic data acquired at a different glacier requires carrying out a new calibration experiment. A potential solution to make empirical models adaptable for different glaciers is to introduce site-specific predictor variables such as source distance, noise level, calving front geometry, water depth, and other seismic wave propagation properties (site and path effects). This approach requires a larger number of calibration data points to prevent overfitting in the regression as well as field campaigns at several glaciers carried out over recording periods longer than 1–2 weeks at each individual site.

## 8   Conclusions

Measuring glacier mass loss through calving continuously and with high temporal resolution is challenging. Seismic and hydroacoustic data recorded at the front of marine-terminating glaciers allow detection of calving signals with second-precision, but require calibration to actual ice volumes. We successfully developed empirical models relating calving signals to volumes at Kronebreen, a fast-flowing tidewater glacier in Northwest Svalbard, based on calving volumes from 10 days of repeat lidar scanning in August and September 2016. In addition to seismic data recorded at three different locations including a permanent station with a more than two decades-long record, we used for the first time a hydroacoustic record. For all records, the amplitude integrated over the calving signals in combination with the maximum amplitude allowed us to model the volumes of individual calving events. While seismic arrays are needed for calving event detection and location, underwater acoustic signals measured on a single sensor showed similar quality and predictor capability for ice volumes as their seismic counterparts and could therefore be used as an complementary record to assess model uncertainty as well as to better understand submarine calving. Reducing seismic calving model uncertainty will benefit from extended calibration data sets at multiple glaciers with shorter lidar scanning intervals and longer recording periods, which better take into account event- and site-dependent variables, and allow improving the matching process between lidar and seismic calving observations. Application of our model showed that the cumulative ice loss estimated from seismic and hydroacoustic data is consistent with measured ice loss in different lidar scanning intervals as well as with frontal ablation rates measured with satellite remote sensing.

Our method offers a promising approach to not only continuously measure calving ice loss with high temporal resolution, but also to indirectly measure frontal glacier melting by evaluating the contribution of dynamic ice loss to the frontal ablation. For the permanent station KBS we model between 15 and 60% of the dynamic ice loss observed close to the calving front since only a fraction of all calving event are detected at 15 km distance depending on seismic noise level. Taking this catalog incompleteness into account allows for long-term monitoring of calving ice loss at Kronebreen. Comparison with total frontal ablation showed that dynamic ice loss contributed 5–30% between 2007 and 2017. This implies a large component of frontal

ablation is melting at the calving face, which we can estimate for Kronebreen in the same period to be roughly $4$–$9\,\mathrm{m\,d^{-1}}$. Future work should consider analysis of higher temporal resolution variability of these estimates which will further unravel the controls on frontal ablation and will be beneficial for realistically modeling ice loss at marine terminating glaciers in relation to both atmospheric and oceanic forcings.

5 *Data availability.* Data of station KBS are freely available through IRIS (Albuquerque Seismological Laboratory (ASL)/USGS, 1988). The seismic record of the temporary network stations will become freely accessible through the Geophysical Instrument Pool Potsdam (GIPP) after 1 October 2020 (Köhler et al., 2019).

## Appendix A: Time-lapse camera image processing

The time-lapse images are first gathered per hour and corrected for lens distortion. The correction is based on the barrel method that follows the equation $R = r\left(a \cdot r^3 + b \cdot r^2 + c \cdot r + d\right)$ where $R$ and $r$ are the distances of the pixels from the center of the original and corrected image, respectively (ImageMagick, Helmut Dersch). The parameters $a$, $b$, and $c$ depend on the camera and lens type and are obtained from the opensource database Lensfun (Table A1). The parameter $d$ is a scaling factor that is greater than 1 and compensates the cropping caused by the correction. Since the position of the calving front is known from satellite images, coordinates of calving events are obtained by projecting the images on the terminus which is assumed to be a straight line for simplification. The geo-referenced images are then rotated to have the waterline horizontal, and applied a crop to only keep the front of the glacier. The parameters for the rotation and crop are manually identified and done every time the camera has moved significantly, usually after each maintenance. To compensate for rapid change in brightness due to environmental conditions or camera settings, we compute a histogram equalization for each color channel separately. This method linearizes the cumulative distribution function of the pixel bins and enhance the contrast of the image. This luminosity correction performs better than stretching pixel values to their maximum range (e.g. normalization). The RGB image is then converted into a grayscale image $Y$ based on a standard Luma compression so that $Y = 0.298839 \cdot R + 0.586811 \cdot G + 0.114350 \cdot B$. The constants are chosen to enhance spatial sensitivity of human vision. Other conversions may be used to enhance colors (i.e. blues), but were not tested here.

The calving detector is developed based on visual characteristics of calving. As the ice breaks off and hits the water, it causes a rapid jet of water and a long-term change in albedo and roughness of the glacier front. The detector aims to identify these temporal variations in pixel brightness that occur along the glacier and above the waterline. The method consists of stacking over time a line of pixel above the front waterline, computing the second derivative in time, apply a low-pass filter and detect events using a threshold value (Fig. A1). The time and location of each calving is retrieved and manually verified using reprocessed gifs of the calving front at these defined time and coordinates.

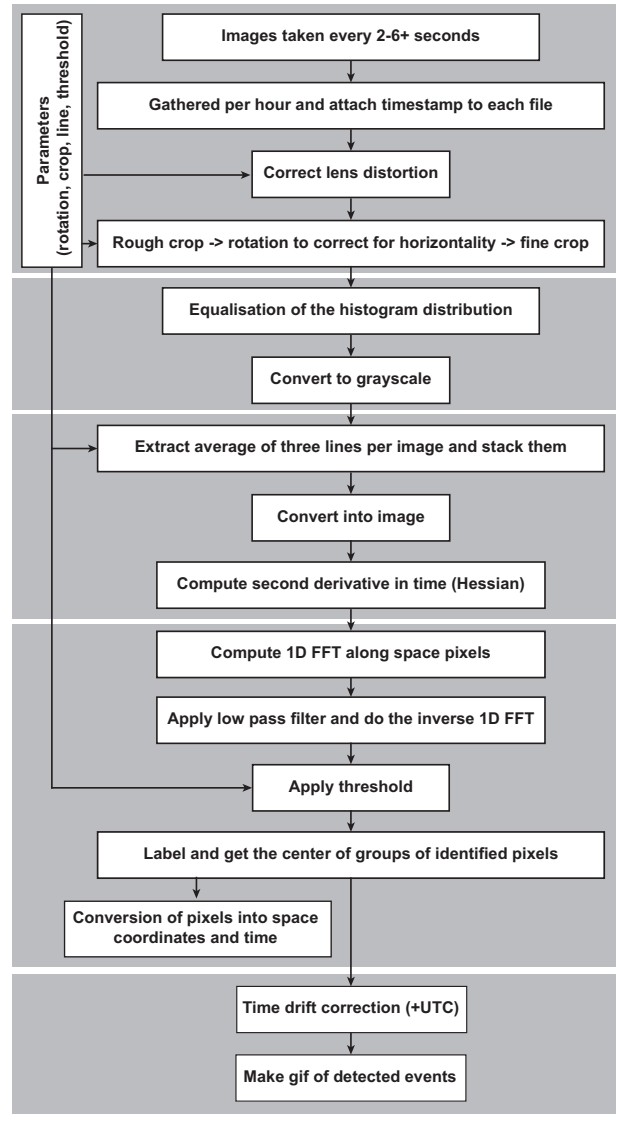

**Figure A1.** Workflow for detecting calving events in the time-lapse image data.

**Table A1.** Camera details and lens distortion parameters

| Camera | | Canon EOS | Canon EOS |
|---|---|---|---|
| Model | | REBEL T3 | 1200D |
| Sensor | mm | 22.2 x 14.7 | 22.3 x 14.9 |
| Image size | px | 4272 x 2848 | 5184 x 3456 |
| Pixel | Mpx | 12.2 | 18.1 |
| JPG size | MB | 2.5 | 6 |
| Lens | | Canon EF-S 18-55mm f/3.5-5.6 IS II | Canon EF-S 18-55mm f/3.5-5.6 III |
| Focal length | | 18 | 18 |
| | a | 0.02504 | 0.00029 |
| Distortion | b | -0.06883 | -0.00065 |
| parameters | c | 0.01502 | -0.04054 |
| | d | 1.15 | 1.15 |

## Appendix B: Repeat lidar scanning

Fig. B1 shows results of lidar DSM differencing for a selected calving event.

## Appendix C: Computation of seismic signal features

Automatic computation of seismic calving signal duration is not trivial because of the complex nature and diversity of waveforms. After testing different approaches, we implemented the following procedure which seems to work best given results from visually inspected events. First, we compute the seismogram envelopes between 1.5 and 5 Hz from a time window which starts 35 s before the event STA/LTA detection time and has a length of 100 s. The envelopes of vertical and both horizontal components are stacked. We start at the detection time and count the number of samples above a defined amplitude threshold, progressing in positive as well as negative time steps. We stop counting if the amplitude falls below that threshold before and after the detection time to avoid previous and later events affecting the duration estimate. The adaptive threshold is computed as follows. First, the lowest value of two mean stacked envelopes computed for 10 s long time segments at the start and the end of the 100 s long time window is chosen. Depending on the station used, this value is then multiplied by a factor between 1.7 and 2.0. To compute the integrated signal amplitude feature, we sum the envelope amplitude of all samples between the obtained start and end time of the signal.

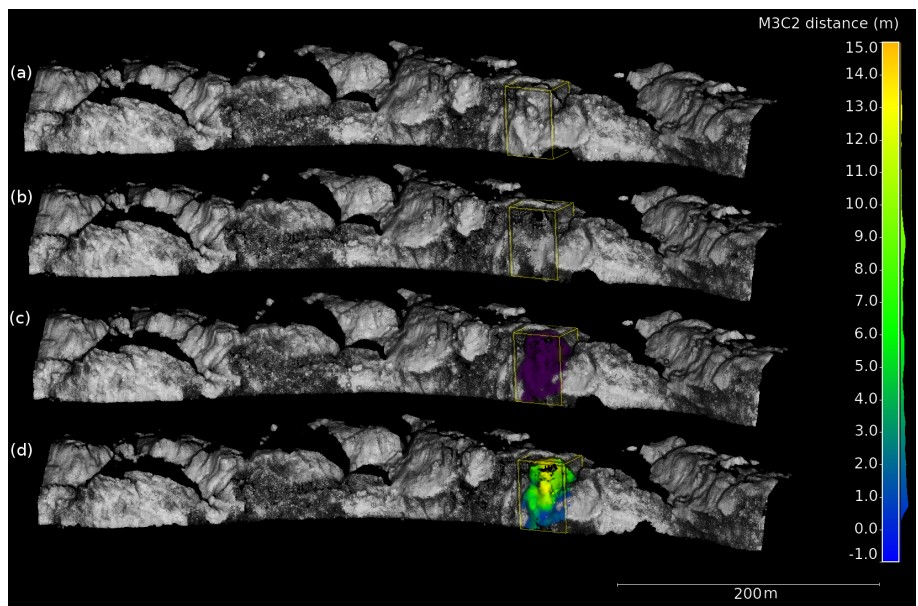

**Figure B1.** Lidar DSM differencing for a calving event on 2016-08-27T09:57:03 (a) point cloud before the event, (b) point cloud after the event, (c) area (purple) of the calving event, (d) calving event depth computed with M3C2 plugin (Lague et al., 2013). A histogram of computed depths is located to the right of the color bar and the calculated calving event volume $V$ is 7046.6 $\pm$ 632.3 m$^3$. The region of interest is marked with a yellow bounding box and the lidar reflection intensity is shown in grayscale on each point cloud (a-d).

## Appendix D: Seismic back-azimuth correction

For seismic signals originating at Kronebreen a discrepancy is observed between back-azimuths measured with FK analysis ($Baz_{seis}$) and real back-azimuths $Baz_{real}$ of calving events visually observed on the time-lapse camera images (Fig. D1b). For correction, a linear dependency of the form $Baz_{real} = a \cdot Baz_{seis} + b$ is assumed. The coefficients $a$ and $b$ are obtained by linear regression using all seismic calving signals confirmed on time-lapse camera images.

## Appendix E: Model calibration

Fig. E1 shows a different visualization of two calibrated models shown in Fig. 3 including uncertainties for lidar measurements.

*Author contributions.* AK and CN initiated the study. AK processed and analyzed the seismic data, built and tested the models, and prepared the manuscript. MP acquired and processed lidar data. PML acquired and processed time-lapse camera images. GB acquired and processed hydroacoustic data. CN was PI of the CalvingSEIS project and planned and organized logistics for the field experiment. CW was responsible for seismic field instrumentation and assisted in the field experiment. All authors contributed to manuscript writing and editing.

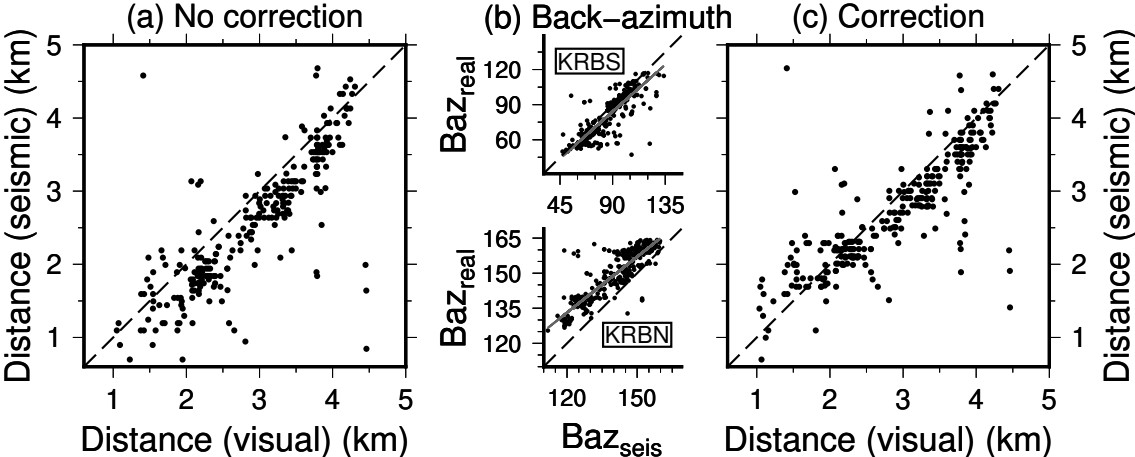

**Figure D1.** Distances from the southern end of the terminus for calving events located from seismic signals and visually observed on time-lapse camera images. (a) No correction. (b) Back-azimuth (degrees from North) of calving signals observed on both seismic arrays vs. visually observed. Gray line shows results of a linear regression. (c) Same as in (a) but applying an empirical correction based on the regression in (b) correcting for observed back-azimuth bias.

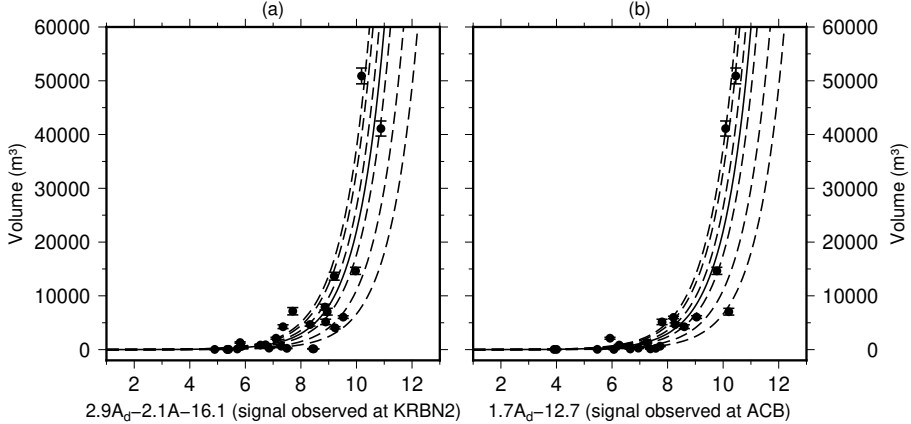

**Figure E1.** Best performing GLMs for KRBN2 and ACB. Symbols show calibration data from lidar measurements with error bars. Solid curve represents the fitted model. Dashed lines show deviations of $\pm 20\%$, $50\%$ and $70\%$ for a given volume. $A_d$ is the logarithm of envelope amplitude integrated over calving signal duration and $A$ the logarithm of maximum envelope amplitude.

*Competing interests.* The authors declare that they have no conflict of interest.

*Acknowledgements.* This study was carried out in the framework of the CalvingSEIS research project funded by the Research Council of Norway (244196/E10). We acknowledge support from SIOS. Seismic instrumentation for temporary arrays was provided by the Geophys-

ical Instrument Pool of GFZ Potsdam, Germany. We use obspy (Beyreuther et al., 2010) for seismic data analysis and statsmodel (Seabold and Perktold, 2010) for GLMs. Figures are produced using GMT (Wessel and Smith, 1998). CN and PL were co-funded by the European Research Council under the European Union's Seventh Framework Programme (FP/2007-2013/ERC; grant agreement no. 320816), and the ESA project Glaciers_cci (4000109873/14/I-NB; 4000127593/19/I-NB). MP acknowledges the support of the Centro de Estudios Científicos (CECs) funded by the Centers of Excellence Base Financing Program of the CONICYT-Chile. We thank Evgeny Podolskiy and an anonymous referee for reviewing the manuscript.

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
