# Peer review of "Contribution of calving to frontal ablation quantified from seismic and hydroacoustic observations calibrated with lidar volume measurements"

_The Cryosphere, 2019_

## Referee Comment (RC1) · Evgeny A. Podolskiy (Referee) · 17 May 2019

The manuscript by Dr. Andreas Kohler et al. presents a new empirical model for estimating calving flux. The model is using seismic and hydro-acoustic records as a proxy for calving, which was calibrated on precise, frequently repeated lidar scans and time-lapse imagery. The uncertainty of the calibrated model is thoroughly assessed before employing it to long-term seismic records of a permanent seismic station (for studying multi-year variations of frontal ablation at the Kronebreen glacier, Svalbard).

The model is based on a short, but very comprehensive field campaign using multiple sensors (seismic, hydro-acoustic, lidar, time-lapse) and diverse methods for signal /

image processing, supported with various statistical models and tests. The manuscript is well written, easy to follow and should be of general and practical interest to the cryospheric community.

As I answer the questions guiding TC referees in their evaluations (https://www.thecryosphere.net/peer\_review/review\_criteria.html), there are only two major aspects I would like to comment on:

1) "Is the description of experiments and calculations sufficiently complete and precise to allow their reproduction by fellow scientists (traceability of results)?"

There are multiple places there descriptions could be improved/completed/corrected as I have specifically listed in MINOR details (e.g., 3 hydrophones deployed, yet locations of only 2 shown; how time-stamping was done underwater to be synced with seismics? hydro-acoustic units are in nm s-1? how signal duration is obtained without "unflag" threshold indicated? and etc.).

Moreover, it would be informative to provide a new figure showing the observed calving statistics in a way similar to recent studies trying to find which law (exp, power, etc.) represents best such distributions as inter-event time and event size? For examples, see Chapuis & Tezlaf, JG, 2014; Petlicki & Kinnard, JG, 2016; Minowa et al., JG, 2018, Minowa et al., EPSL, 2019. I guess, such plots will also better expose completeness issues of the dataset.

2) "Do the authors give proper credit to related work and clearly indicate their own new/original contribution?"

To my knowledge, the contribution is indeed novel. However, as it comes to Introduction and Discussion, I would like to point out the following. There are more recent studies than those "qualitatively estimating the relation between ... calving ... and ...volume" [p.14, Line 3]. For example, continuous high-res. calving records were recently obtained by using another calibrated model, which is linking water-surface waves TCD
with time-lapse- or UAV- derived calved ice volumes (Minowa et al., J. Glaciol., 2018 & EPSL, 2019). The latter studies are of a high relevance to presented here analysis, because they report very comparable findings (for example, the calved volume is up to 105 m3 and contribution of calving to frontal ablation is 20% at another arctic tide-water glacier, while subaqueous calving is negligible).

Also, calibrated seismic models for a particular glacier/site are not "The only successful approach so far" [p. 2, line 11]. Because recently there was a progress in developing seismo-mechanical approaches, for example, for glacial earthquakes by Sergeant et al. (Ann. Glaciol. 2019). Moreover, it is not correct to associate glacial earthquakes with tabular icebergs in Greenland; those are usually nontabular.

**MINOR**

p.1 affiliation #3: missing city, country

p.2, line 1: please, provide literature examples to "radar and liar surveys"

p.2, line 5: "Calving events are also successfully detected from  $\dots$ "/  $\dots$  and water surface (or tsunami) waves (Minowa et al., 2018 & 2019)

p.2, line 6: "Seismic and hydroacoustic methods have the advantage to produce continuous calving records..."/ -> please consider "Seismic, hydroacoustic and water-wave methods"

p.2, line 10: for glacial earthquake signals from tabular iceberg calving such as observed in Greenland (Murray et al., 2015)

-> it is misleading to use "tabular iceberg" here, because usually tabular icebergs do not capsize and do not generate significant glacial earthquakes (Sergeant et al., 2019). Consider re-writing as:

"for glacial earthquake signals from buoyancy-driven nontabular iceberg calving such as observed in Greenland (Murray et al., 2015; Sergeant et al., 2019)."
p.2, line 11: iceberg impact on the sea surface -> please consider re-writing serac impact on the sea/lake surface

p.2, line 11: "The only successful approach so far" -> see my comment above (about Sergeant et al. 2019)

p.2, line 27: Ny Alesund or Ny-Alesund? -> please, use consistent dash/no dash through out the manuscript.

p.2, line 29: with more than 1 km -> {for} more than 1 km?

p.2, line 31: (Nuth et al., in preparation) ->Please, note that according to TC guide for authors [Works "submitted to", "in preparation", "in review", or only available as preprint should also be included in the reference list.]

p.3, lines 8-16:

->3 hydrophones deployed, yet locations of only 2 shown. I could not follow the destiny of the third recorder.

->please, explain how time-stamping of the hydroacoustic data and possible time drift was dealt with for comparing with seismic data?

The recording unit was recovered -> The recording units were recovered

p.5, line 9: scans of acquired from ->{of} could be omitted?

p.5, line 10: to the west of the main camp. ->Does the reader know where is the main camp in the first place?

p.5, line 14: mismatches ... was removed ->were removed

p.20, lines 21-22: Calving event scars have low reflectance in near infrared part of spectrum -> is it your finding? Perhaps, a reference should be helpful here?

p.20, lines 26: point cloud differences ... is added ->are added

TCD
Figure 2: hydroacoustic pressure has nm s-1 units?

lambda/4 criterion: -> please, elaborate or reference this criterion

p.7 line 8: when for example ocean -> when, for example, ocean

p.7 line 22: signal duration ->please explain how do you define the duration - otherwise it remains unclear because earlier mentioned STA/LTA was applied to array and did not show any flag-off threshold.

Figure 3: I suggest to explain abbreviation GLM because the reader sees it before reading the text.

here, and in Fig. 5, it would not hurt to explain in the captions the dashed 1:1 lines.

p.9 line 13: using the deviance, the common -> using the deviance, d, the common

p.9 line 15: Perhaps, you could add that null deviance is the same for all models of each station?

p.9, line 26: . i.e., -> For example,

p.10, lines 16-17: Additionally to total ice volume, I suggest to show how much volume per day does it correspond to - for easier comparison with other studies.

p.10, line 18: The volumes ... lays -> lay

Table 1: KRBN -> KRBN2 ?

KRBS -> KRBS3 ?

Figure 6, caption: Right panels shows -> panel (there is only one)

consider re-writing as: Right panel\_ shows {the corresponding} box plot{s} for ...

p.14, beginning of Discussion: see my previous comment that there are more recent calving flux estimation studies based on "tsunami" wave monitoring. As you would see from the following 5 remarks below, I believe, it is highly relevant work to your study.
p. 15 - as in other places, line numbering has collapsed here, so I just cite the place: "dynamic ice loss to frontal ablation at Kronebreen of 5-30%" ->Similarly, Minowa et al. (2019) finds 20% at another tide-water glacier.

p.15, lines 8-9: It is well-known that also submarine calving generates underwater acoustic and weak seismic signals. -> as well as water surface waves (Minowa et al., 2018).

p.15, lines 18-19: 97% of all events occurred subaerially ->Similarly, Minowa et al. (2018) finds 98%.

p.15, lines 18-19: Another possible solution is to ... ->use UAV-derived DEM (Minowa et al., 2019)

p.16, line 1: please, consider giving an example to previous efforts like this suggestion of yours: to measure calving areas from time-lapse images (e.g., Minowa et al., 2018)...

p. 16: "calving in Greenland and Antarctica is dominated by the breakup of large tabular icebergs representing a different type of seismic source mechanism (Murray et al., 2015)."

-> in line with my previous comment, it should not be tabular here for Greenland. Moreover, Murray's paper is not about Antarctica, you could refer to Chen et al. (JGR, 2011) for this. Therefore, please, consider correcting as:

"calving in Antarctica and Greenland is dominated by the breakup of large tabular and nontabular icebergs, respectively, representing different types of seismic source mechanisms (Murray et al., 2015; Chen et al., 2011; Sergeant et al., 2019)."

p.16, line 9: a different or more generalized approach. -> (Sergeant et al., 2019)

p.16, line 15: carried out over longer recording periods than 1–2 weeks -> carried out over recording periods longer than 1–2 weeks

p.16, line 26: to asses model -> to asses{s}
p.17, line 6: contributed with 5-30% ->do you need [with] here?

p. 18, line ? - line numbering collapsed here:

As the ice breaks off and hit -> hits

variations in pixel brightness that occurs ->variations ... that occur

location and coordinates of calving from gifs - I am not sure how do you get location from non-geo-referenced (?) images. Manually? Elaborate, please.

Figure B1:

Grey line is results -> Grey line shows results

I also note that you keep using [grey] and [gray], why not to stick to just one spelling?

---

## Referee Comment (RC3) · Anonymous Referee #2 · 21 Aug 2019

Review: Kohler et al., 2019 TCD

Dear colleagues,

The manuscript "Contribution of calving to frontal ablation quantified from seismic and hydroacoustic observations calibrated with lidar volume measurements" by Kohler et al. presents nice and interesting results. It investigates frontal ablation based on comprehensive field dataset and suggests a possible large contribution of submarine melting to frontal ablation. This is a timely topic in the community and relevant for the journal. The manuscript is well written and easy to understand. After some enhancements of results and discussion, I recommend publication.

[Figure]

Sincerely,

General comments 1. I understand that the accurate estimation of iceberg volume from lidar is one of the foundations of the study. Therefore, I think you need to present a figure showing DSMs differences in the article. Perhaps, you could show such plot above Figure 2? Also, I wonder what the uncertainty range of the calculated iceberg volume from the lidar datasets is.

2. I am curious to see the cumulative distribution function of modeled iceberg volumes, which you could investigate the completeness of iceberg volume from their distribution. Perhaps, you could show such plots beside Figure 4?

Specific comments Page 1: author name: remove space between the author names and superscripts numbers.

Page 1, line 9: 18-30% -> use en-dash instead of hyphen. I found the same typo many times later in the manuscript. You need to correct them accordingly.

Page 2, line 14: Yahtse glacier -> Yahtse "G"lacier

Page 2, Study site: It would be worth to introduce a bit more info about the glacier for readers who are not familiar with the glacier. For instance, how fast the glacier? How wide the ice front? How deep the fjord near the ice front? How warm or cold the fjord? I believe that this information would be useful later to think of the relevance of the method to apply to other regions.

Page 3, line 10: use minus but hyphen: -166 $\pm$ 1

Page 3, line 27: Only a few percent. . . -> I prefer to know an explicit number. Also how big was submarine calving iceberg at the glacier? Could be much bigger than subaerial calving?

Page 4, Repeat lidar scanning: I'm not familiar with lidar scanning but does lidar signal penetrate ice? What is the uncertainty of the calculated iceberg volumes, after all?

Page 6, Figure 2: I want to see a plot of DSMs differentiation of the corresponding calving event presented in Figure 2. Is it possible to show such plot above Figure 2?

Page 7, line 9: Were calving events uniformly distributed along the ice front? Did you find any spatial distribution of the located calving events?

Page 8, line 8–9: How did you find real-world coordinates of calving events from time-lapse images?

Page 9, Equation 2: Would be better to make the outermost parentheses lager for better readability.

Page 10, line 27: I'm curious to see CDF of the modeled iceberg volume so that we can learn which size of calving is missing.

Page 13, line 28: insert space before and after +/- sign.

Page 15, line 5: It the inferred submarine melt rate is consistent with the recently reported submarine melt rate at the glacier (Schild et al., 2018; Holmes et al., 2019)?

Page 15, line 10–11: Did you confirm the spectral difference between submarine and subaerial calving on hydroacoustic signals at the glacier?

Page 16, line 5: "South America" -> "Patagonia" makes more sense to me.

Page 16, line 24: Apart from the novelty to use hydroacoustic signals, did you find any advantage to use hydroacoustic signals with seismic signals? Perhaps, you may have a chance to include submarine calving to the model by using hydroacoustic signal somehow?

Figure A1. How did you convert pixels into the real-world coordinate?

Figure B1. I am somewhat unhappy with the units of the plot. Consider using meter or kilometer instead of latitude.

References Holmes, F. A., Kirchner, N., Kuttenkeuler, J., Krützfeldt, J., & Noormets,

[Figure]

R. (2019). Relating ocean temperatures to frontal ablation rates at Svalbard tidewater glaciers: Insights from glacier proximal datasets. Scientific Reports, 9(1), 1–11. https://doi.org/10.1038/s41598-019-45077-3

Schild, K. M., Renshaw, C. E., Benn, D. I., Luckman, A., Hawley, R. L., How, P., . . . Hulton, N. R. J. (2018). Glacier Calving Rates Due to Subglacial Discharge, Fjord Circulation, and Free Convection. Journal of Geophysical Research: Earth Surface, 123(9), 2189–2204. https://doi.org/10.1029/2017JF004520

---

## Author Comment (AC1) · 26 Sep 2019

We very much appreciate the very detailed and thorough review provided by Evgeny Podolskiy. We especially value the feedback concerning the missing literature references. A response to all points raised by the reviewer can be found attached to this comment along with a revised, preliminary version of the manuscript highlighting all modifications. Any further feedback is appreciated.

Please also note the supplement to this comment:
https://www.the-cryosphere-discuss.net/tc-2019-75/tc-2019-75-AC1-supplement.pdf

---

## Author Comment (AC2) · 26 Sep 2019

We very much appreciate this very helpful review that helped us clarifying issues and adding missing information. A response to all points raised by the reviewer can be found attached to this comment along with a revised, preliminary version of the manuscript highlighting all modifications. Any further feedback is appreciated.

Please also note the supplement to this comment:
https://www.the-cryosphere-discuss.net/tc-2019-75/tc-2019-75-AC2-supplement.pdf

[Figure]

**Supplement:**

**Response to Referee 2**

We would like to thank the reviewer for his very helpful comments and suggestions. We responded to all points raised by the reviewer below and attached a revised version of the manuscript highlighting all modifications.

General comments

1. I understand that the accurate estimation of iceberg volume from lidar is one of the foundations of the study. Therefore, I think you need to present a figure showing DSMs differences in the article. Perhaps, you could show such plot above Figure 2? Also, I wonder what the uncertainty range of the calculated iceberg volume from the lidar datasets is.

We agree that a figure demonstrating and showing the results of lidar scanning would be helpful. We added a new figure as supporting material which visualizes a measured calving event volume. Furthermore, we added measurement uncertainties of lidar volumes in Fig. C1 and Fig.5.

2. I am curious to see the cumulative distribution function of modeled iceberg volumes, which you could investigate the completeness of iceberg volume from their distribution. Perhaps, you could show such plots beside Figure 4?

We agree that such a plot would be beneficial (also suggested by referee 1). We created an additional figure showing the calving volume and inter-event interval statistics, as well as fits of distribution models. We added a paragraph briefly describing and discussing the results.

Specific comments

Page 1: author name: remove space between the author names and superscripts numbers.

Removed.

Page 1, line 9: 18-30% -> use en-dash instead of hyphen. I found the same typo many times later in the manuscript. You need to correct them accordingly.

We scanned through the manuscript and corrected. Thanks for recognizing this typo.

Page 2, line 14: Yahtse glacier -> Yahtse "G"lacier

Corrected.

Page 2, Study site: It would be worth to introduce a bit more info about the glacier for readers who are not familiar with the glacier. For instance, how fast the glacier? How wide the ice front? How deep the fjord near the ice front? How warm or cold the fjord? I believe that this information would be useful later to think of the relevance of the method to apply to other regions.

We added more information about terminus size, glacier flow velocity and fjord temperature as suggested.

Page 3, line 10: use minus but hyphen: -166±1

Corrected.

Page 3, line 27: Only a few percent...-> I prefer to know an explicit number. Also how big was submarine calving iceberg at the glacier? Could be much bigger than subaerial calving?

Since sub-marine calving is more difficult to detect in time-lapse images, we cannot give a certain number here. However, our estimate based on visually screened images is that sub-marine calving contributes with less than 5% (Information added). Size of sub-marine calving is also difficult to assess because a big part remains submerged in the fjord. From the few observed events, we estimate that sizes are not bigger than the largest subaerial events we observe.

Page 4, Repeat lidar scanning: I'm not familiar with lidar scanning but does lidar signal penetrate ice? What is the uncertainty of the calculated iceberg volumes, after all?

We added uncertainties (error bars) for individual events in Fig. C1 and for cumulative volumes in Fig. 5 as well as in Table 1. According to Deems and others (2013) lidar penetration depth is limited to few cm or less. There is no study of

glacial ice, but for clean ice one would expect penetration depth in the order of 4-5 cm based on experiments of Grenfell and Perovich (1981). This depth should further decrease for glacial ice containing air bubbles. Some general discussion on this subject can be found in Podgórski and others (2018), but certainly it is an interesting subject to further explore in future studies. Even if the laser penetrated more than the values given above, let's say 10 cm, this would be a systematic error consistent throughout the dataset and would be canceled during point cloud differencing (both point clouds would have the same bias). Hence, laser penetration depth is not influencing the final result.

Deems J, Painter T, and Finnegan D (2013). Lidar measurement of snow depth: A review. Journal of Glaciology, 59(215), 467-479. doi:10.3189/2013JoG12J154

Grenfell TC and Perovich DK (1981) Radiation absorption coefficients of polycrystalline ice from 400–1400 nm. Journal of Geophysical Research, 86(C8), 7447. doi:10.1029/jc086ic08p0744781

Page 6, Figure 2: I want to see a plot of DSMs differentiation of the corresponding calving event presented in Figure 2. Is it possible to show such plot above Figure 2?

We added such a plot as supplementary material. We hope this is sufficient.

Page 7, line 9: Were calving events uniformly distributed along the ice front? Did you find any spatial distribution of the located calving events?

Fig. 1 shows the spatial distribution of located seismic calving signals and a histogram of time-lapse camera detected events along the terminus. Results show that indeed some parts are more active. We added a sentence and a reference to the figure.

Page 8, line 8–9: How did you find real-world coordinates of calving events from time-lapse images?

We forgot to mention that geo-referencing is included in the processing. Since we know the position of the calving front from satellite images and location of the camera, we can obtain coordinates from the time lapse images by projecting the image on the terminus which is assumed to be a straight line for simplification. Missing information was added.

Page 9, Equation 2: Would be better to make the outermost parentheses lager for better readability.

Changed.

Page 10, line 27: I'm curious to see CDF of the modeled iceberg volume so that we can learn which size of calving is missing.

We added a figure showing the cumulative volume distribution and discuss the results in a new paragraph.

Page 13, line 28: insert space before and after +/- sign.

Corrected.

Page 15, line 5: Is the inferred submarine melt rate is consistent with the recently reported submarine melt rate at the glacier (Schild et al., 2018; Holmes et al., 2019)?

Thank you for pointing us to these studies which are very relevant to add as references. While comparing results, we found a mistake in our melt rate calculation and now use the correct cross-sectional area of the glacier terminus (about 0.25 km2). With estimated melt volumes of 0.4-0.8 km3/a (see Fig 7), we obtain now melt rates about 4-9 m/d.

Schild et al. (2018) reported melt rates between 0.1 and 6.8 m/d between April and October 2016 which is of the same order as our average rate estimates for the whole time period of 11 years. The modeled melt rate at Kronebreen according to Holmes et al. (Fig. 3 in that paper) varies seasonally between 0.5 and 2.0 m3/s which corresponds to 0.015-0.06 km3/a. This is one order of magnitude lower than our estimate. However, Holmes et al. point out that their melt rates are probably underestimated. The melt rate is modeled from fjord temperatures and does not consider melting from plumes, despite evidence for those. Therefore, the authors found that their result only account for 25.6% of the frontal ablation. The higher contribution to total frontal ice loss that we found is also similar to those for other Arctic tidewater glaciers (e.g. Miowa et. al., 2019) which further adds confidence that these estimates may be more realistic.
We added the comparison to those studies in the discussion section.

Page 15, line 10–11: Did you confirm the spectral difference between submarine and subaerial calving on hydroacoustic signals at the glacier?

We did not systematically study differences in spectral characteristics of hydroacoustic calving signals in this study. This is mainly because of the low number of identified submarine events and the focus of this paper on calving quantification. Furthermore, the long-term calving record is based on seismic data only. However, we agree that our data set can and should be used to study in more detail differences of underwater signals generated by different calving styles, similar to the work done at Hansbreen. We added this recommendation in the discussion.

Page 16, line 5: "South America" -> "Patagonia" makes more sense to me.

Changed.

Page 16, line 24: Apart from the novelty to use hydroacoustic signals, did you find any advantage to use hydroacoustic signals with seismic signals? Perhaps, you may have a chance to include submarine calving to the model by using hydroacoustic signal somehow?

We agree that hydroacoustic records may help to better capture ice loss through submarine calving, although its contribution seems to be low. Furthermore, for logistical reasons (e.g., accessibility) an hydroacoustic instrument may be the only sensor available for recording calving at remote glaciers. In our study we demonstrate that such a record can be used in the same way as seismic data to quantify ice loss. As written in the conclusions, underwater acoustic signals measured on a single sensor showed similar quality and predictor capability for ice volumes as their seismic counterparts and could therefore be used as a complementary record to assess model uncertainty.

Figure A1. How did you convert pixels into the real-world coordinate?

We added the missing information.

Figure B1. I am somewhat unhappy with the units of the plot. Consider using meter or kilometer instead of latitude.

We changed to km.

[revised manuscript text omitted]

---

## Author Comment (AC3) · 26 Sep 2019

We very much appreciate the very detailed and thorough review provided by Evgeny Podolskiy. We especially value the feedback concerning the missing literature references. A response to all points raised by the reviewer can be found attached to this comment along with a revised, preliminary version of the manuscript highlighting all modifications. Any further feedback is appreciated.

Please also note the supplement to this comment:
https://www.the-cryosphere-discuss.net/tc-2019-75/tc-2019-75-AC3-supplement.pdf

**Supplement:**

**Response to Evgeny A. Podolskiy (Referee 1)**

First of all, we would like to thank Evgeny Podolskiy for again being willing to review a paper of ours within a short time period. We appreciate his detailed and thorough review of our manuscript. We responded to all points raised by the reviewer below and attached a revised version of the manuscript highlighting all modifications.

1) "Is the description of experiments and calculations sufficiently complete and precise to allow their reproduction by fellow scientists (traceability of results)?"

There are multiple places where descriptions could be improved/completed/corrected as I have specifically listed in MINOR details (e.g., 3 hydrophones deployed, yet locations of only 2 shown; how time-stamping was done underwater to be synced with seismics? hydro-acoustic units are in nm s-1? how signal duration is obtained without "unflag" threshold indicated? and etc.).

We addressed all these issues below.

Moreover, it would be informative to provide a new figure showing the observed calving statistics in a way similar to recent studies trying to find which law (exp, power, etc.) represents best such distributions as inter-event time and event size? For examples, see Chapuis & Tezlaf, JG, 2014; Petlicki & Kinnard, JG, 2016; Minowa et al., JG, 2018, Minowa et al., EPSL, 2019. I guess, such plots will also better expose completeness issues of the dataset.

We agree that this would be a very useful plot. We added a new figure showing the calving volume and inter-event interval statistics, as well as fits of distribution models. We added a paragraph briefly describing and discussing the results.

2) "Do the authors give proper credit to related work and clearly indicate their own new/original contribution?"

To my knowledge, the contribution is indeed novel. However, as it comes to Introduction and Discussion, I would like to point out the following. There are more recent studies than those "qualitatively estimating the relation between . . . calving . . . and. . .volume" [p.14, Line 3]. For example, continuous high-res. calving records were recently obtained by using another calibrated model, which is linking water-surface waves with time-lapse- or UAV- derived calved ice volumes (Minowa et al., J. Glaciol., 2018 & EPSL, 2019). The latter studies are of a high relevance to presented here analysis, because they report very comparable findings (for example, the calved volume is up to 10^5 m3 and contribution of calving to frontal ablation is 20% at another arctic tide-water glacier, while subaqueous calving is negligible).

We absolutely agree that both recent papers are very relevant for our study. In fact, I found these papers shortly after we submitted our manuscript, and my intention was to include them in the revised version anyway. Both papers are now referred to in the introduction as well as several times in the discussion as suggested.

Also, calibrated seismic models for a particular glacier/site are not "The only successful approach so far" [p. 2, line 11]. Because recently there was a progress in developing seismo-mechanical approaches, for example, for glacial earthquakes by Sergeant et al. (Ann. Glaciol. 2019). Moreover, it is not correct to associate glacial earthquakes with tabular icebergs in Greenland; those are usually nontabular.

Also, in this case, timing was the issue that this paper was not included yet. I had planned to add this reference in the revised version. As to the second point, we agree that our description is misleading. We wanted to emphasize that the source mechanism generating glacier earthquakes in Greenland (capsizing, full-glacier-thickness icebergs) which do not seem to occur in Svalbard, is different from the one generating iceberg impact signals that we measure in Svalbard (which of course also occur in Greenland). The reason is most likely that Greenland outlet glaciers are much larger. We clarified and removed "tabular" which we agree is not correct in this context.

MINOR
p.1 affiliation #3: missing city, country

Added.

p.2, line 1: please, provide literature examples to "radar and lidar surveys"

We added references for both methods.

p.2, line 5: "Calving events are also successfully detected from . . ."/ . . . and water surface (or tsunami) waves (Minowa et al., 2018 & 2019)

p.2, line 6: "Seismic and hydroacoustic methods have the advantage to produce continuous calving records. . ."/ -> please consider "Seismic, hydroacoustic and water-wave methods"

We added monitoring calving through tsunamis (method and references).

p.2, line 10: for glacial earthquake signals from tabular iceberg calving such as observed in Greenland (Murray et al., 2015) -> it is misleading to use "tabular iceberg" here, because usually tabular icebergs do not capsize and do not generate significant glacial earthquakes (Sergeant et al., 2019).
Consider re-writing as: "for glacial earthquake signals from buoyancy-driven nontabular iceberg calving such as observed in Greenland (Murray et al., 2015; Sergeant et al., 2019)."

We rephrased. See also our answer above.

p.2, line 11: iceberg impact on the sea surface -> please consider re-writing serac impact on the sea/lake surface

We agree that "lake surface" should be added. However, we believe that these signals should not be restricted to serac impacts, but also include topple and drop calving styles (according to Fig.2 in Minowa et al. 2018). We rephrased.

p.2, line 11: "The only successful approach so far" -> see my comment above (about Sergeant et al. 2019)

We rephrased this section to include the methods of Sergeant et al. 2019 and Minowa et al. 2019.

p.2, line 27: Ny Alesund or Ny-Alesund? -> please, use consistent dash/no dash through out the manuscript.

Corrected.

p.2, line 29: with more than 1 km -> {for} more than 1 km?

Corrected.

p.2, line 31: (Nuth et al., in preparation) ->Please, note that according to TC guide for authors [Works "submitted to", "in preparation", "in review", or only available as preprint should also be included in the reference list.]

Added to reference list.

p.3, lines 8-16:
->3 hydrophones deployed, yet locations of only 2 shown. I could not follow the destiny of the third recorder.
->please, explain how time-stamping of the hydroacoustic data and possible time drift was dealt with for comparing with seismic data?

The clock of the recorder was synchronized with GPS time at deployment. The clock drift measured after recovery of sensor ACB was about 10 s. We corrected for the time drift accordingly assuming a linear trend. Information was added. The third recorder was located outside the map area and was not used in this study. We removed this information to avoid confusion.

The recording unit was recovered -> The recording units were recovered

Corrected.

p.5, line 9: scans of acquired from ->{of} could be omitted?

Removed "of".

p.5, line 10: to the west of the main camp. ->Does the reader know where is the main camp in the first place?

The main camp is equivalent with the camera and lidar position. We added this information.

p.5, line 14: mismatches . . . was removed ->were removed

Corrected.

p.20, lines 21-22: Calving event scars have low reflectance in near infrared part of spectrum -> is it your finding Perhaps, a reference should be helpful here?

We added a reference.

p.20, lines 26: point cloud differences . . . is added ->are added

Corrected.

Figure 2: hydroacoustic pressure has nm s-1 units?

Thank you for noticing this plotting error. In contrast to seismic data, we did not convert hydroacoustic data into physical units (i.e., pressure), but used the raw output of the digitizer. The unit is therefore digitizer counts. Corrected.

lambda/4 criterion: -> please, elaborate or reference this criterion

Here, we simply adapt the Rayleigh criterion which is for example used in seismic processing to define the limit of resolution for two signals originating from two refractors and which is approximately lambda/4. We use it as an approximation for the minimum distance between two seismic sources that can be separated for a given wavelength. We rephrased.

p.7 line 8: when for example ocean -> when, for example, ocean

Corrected.

p.7 line 22: signal duration ->please explain how do you define the duration – otherwise it remains unclear because earlier mentioned STA/LTA was applied to array and did not show any flag-off threshold.

We added a detailed description of how the signal duration was computed in the Appendix.

Figure 3: I suggest to explain abbreviation GLM because the reader sees it before reading the text. here, and in Fig. 5, it would not hurt to explain in the captions the dashed 1:1 lines.

Figure caption updated.

p.9 line 13: using the deviance, the common -> using the deviance, d, the common
p.9 line 15: Perhaps, you could add that null deviance is the same for all models of each station?
p.9, line 26: . i.e., -> For example,

Done.

p.10, lines 16-17: Additionally to total ice volume, I suggest to show how much volume per day does it correspond to - for easier comparison with other studies.

We added the average daily rates in the text for comparison with other studies.

p.10, line 18: The volumes . . . lays -> lay
Table 1: KRBN -> KRBN2 ? KRBS -> KRBS3 ?
Figure 6, caption: Right panels shows -> panel (there is only one) consider re-writing as: Right panel_ shows {the corresponding} box plot{s} for . . .

Corrected.

p.14, beginning of Discussion: see my previous comment that there are more recent calving flux estimation studies based on "tsunami" wave monitoring. As you would see from the following 5 remarks below, I believe, it is highly relevant work to your study.
p. 15 - as in other places, line numbering has collapsed here, so I just cite the place: "dynamic ice loss to frontal ablation at Kronebreen of 5-30%" ->Similarly, Minowa et al. (2019) finds 20% at another tide-water glacier.
p.15, lines 8-9: It is well-known that also submarine calving generates underwater acoustic and weak seismic signals. -> as well as water surface waves (Minowa et al.,2018).
p.15, lines 18-19: 97% of all events occurred subaerially ->Similarly, Minowa et al. (2018) finds 98%.

p.15, lines 18-19: Another possible solution is to . . . ->use UAV-derived DEM (Minowa et al., 2019)

p.16, line 1: please, consider giving an example to previous efforts like this suggestion of yours: to measure calving areas from time-lapse images (e.g., Minowa et al., 2018)..

p. 16: "calving in Greenland and Antarctica is dominated by the breakup of large tabular icebergs representing a different type of seismic source mechanism (Murray et al., 2015)."

-> in line with my previous comment, it should not be tabular here for Greenland. Moreover, Murray's paper is not about Antarctica, you could refer to Chen et al. (JGR, 2011) for this. Therefore, please, consider correcting as: "calving in Antarctica and Greenland is dominated by the breakup of large tabular and nontabular icebergs, respectively, representing different types of seismic source mechanisms (Murray et al., 2015; Chen et al., 2011; Sergeant et al., 2019)."

p.16, line 9: a different or more generalized approach. -> (Sergeant et al., 2019)

We added the missing references as suggested and rephrased. Thank you again for pointing us to these studies.

p.16, line 15: carried out over longer recording periods than 1–2 weeks -> carried out over recording periods longer than 1–2 weeks

Corrected

p.16, line 26: to asses model -> to asses{s}

Corrected

p.17, line 6: contributed with 5-30% ->do you need [with] here?

Corrected

p. 18, line ? - line numbering collapsed here:

As the ice breaks off and hit -> hits
variations in pixel brightness that occurs ->variations . . . that occur

Corrected

location and coordinates of calving from gifs - I am not sure how do you get location from non-geo-referenced (?) images. Manually? Elaborate, please.

We forgot to mention that geo-referencing is included in the processing. Since we know the position of the calving front from satellite images and location of the camera, we can obtain coordinates from the time lapse images by projecting the image on the terminus which is assumed to be a straight line for simplification. Missing information was added.

Figure B1:
Grey line is results -> Grey line shows results
I also note that you keep using [grey] and [gray], why not to stick to just one spelling?

Corrected

[revised manuscript text omitted]